# Improved Regret Bounds for Tracking Experts with Memory

**James Robinson**
Department of Computer Science
University College London
London
United Kingdom
j.robinson@cs.ucl.ac.uk

**Mark Herbster**
Department of Computer Science
University College London
London
United Kingdom
m.herbster@cs.ucl.ac.uk

## Abstract

We address the problem of sequential *prediction with expert advice* in a non-stationary environment with long-term memory guarantees in the sense of Bousquet and Warmuth [4]. We give a linear-time algorithm that improves on the best known regret bounds [27]. This algorithm incorporates a relative entropy projection step. This projection is advantageous over previous weight-sharing approaches in that weight updates may come with implicit costs as in for example portfolio optimization. We give an algorithm to compute this projection step in linear time, which may be of independent interest.

## 1 Introduction

We consider the classic problem of online prediction with expert advice [35] in a non-stationary environment. In this model `nature` sequentially generates outcomes which the `learner` attempts to predict. Before making each prediction, the `learner` listens to a set of $n$ experts who each make their own predictions. The `learner` bases its prediction on the advice of the experts. After the prediction is made and the true outcome is revealed by `nature`, the accuracies of the `learner`'s prediction and the expert predictions are measured by a loss function. The `learner` receives information on all expert losses on each trial. We make no distributional assumptions about the outcomes generated, indeed `nature` may be assumed to be adversarial. The goal of the `learner` is to predict well relative to a predetermined comparison class of predictors, in this case the set of experts themselves. Unlike the standard regret model, where the `learner`'s performance is compared to the single best predictor in hindsight, our aim is for the `learner` to predict well relative to a sequence of comparison predictors. That is, "switches" occur in the data sequence and different experts are assumed to predict well at different times.

In this work our focus is on the case when this sequence consists of a few unique predictors relative to the number of switches. Thus most switches return to a previously "good" expert, and a `learner` that can exploit this fact by "remembering" the past can adapt more quickly than a `learner` who has no memory and must re-learn the experts after every switch. The problem of switching with memory in online learning is part of a much broader and fundamental problem in machine learning: how a system can adapt to new information yet retain knowledge of the past. This is an area of research in many fields, including for example, catastrophic forgetting in artificial neural networks [11, 36].

**Contributions.** In this paper we present an $\mathcal{O}(n)$-time per trial projection-based algorithm for which we prove the best known regret bound for tracking experts with memory. Our projection-based algorithm is intimately related to a more traditional "weight-sharing" algorithm, which we show is a new method for *Mixing Past Posteriors* (MPP) [4]. We show that surprisingly this method

35th Conference on Neural Information Processing Systems (NeurIPS 2021).

corresponds to the algorithm with the previous best known regret bound for this problem [27]. We also give an efficient $\mathcal{O}(n)$-time algorithm for computing exact relative entropy projection onto a simplex with non-uniform (lower) box constraints. Finally, we provide a guarantee which favors projection-based updates over weight-sharing updates when updating weights may incur costs.

The paper is organized as follows. We first introduce the model and discuss related work, giving a detailed overview of the previous results on which we improve. In Section 3 we give our main results, a regret bound which holds for two algorithms, and an algorithm to compute relative entropy projection with non-uniform lower box constraints in linear time. In Section 4 we derive a new "geometric-decay" method for MPP, and show the correspondence to the current best known algorithm [27]. We give a few concluding remarks in Section 5. All proofs are contained in the appendices.

## 1.1 Preliminaries

We first introduce notation. Let $\Delta_n := \{\boldsymbol{u} \in [0,1]^n : \|\boldsymbol{u}\|_1 = 1\}$ be the $(n-1)$-dimensional probability simplex. Let $\Delta_n^\alpha := \{\boldsymbol{u} \in [0,\alpha]^n : \|\boldsymbol{u}\|_1 = \alpha\}$ be a scaled simplex. Let $\mathbf{1}$ denote the vector $(1, \ldots, 1)$ and $\mathbf{0}$ denote the vector $(0, \ldots, 0)$. Let $\boldsymbol{e}_i$ denote the $i^{th}$ standard basis vector. We define $D(\boldsymbol{u}, \boldsymbol{w}) := \sum_{i=1}^n u_i \log \frac{u_i}{w_i}$ to be the relative entropy between $\boldsymbol{u}$ and $\boldsymbol{w}$. We denote component-wise multiplication as $\boldsymbol{u} \odot \boldsymbol{w} := (u_1 w_1, \ldots, u_n w_n)$. For $p \in [0,1]$ we define $\mathcal{H}(p) := -p \ln p - (1-p) \ln (1-p)$ to be the binary entropy of $p$, using the convention that $0 \ln 0 = 0$. We define ri $S$ to be the relative interior of the set $S$. For any positive integer $n$ we define $[n] := \{1, \ldots, n\}$. We overload notation such that $[\texttt{pred}]$ is equal to 1 if the predicate $\texttt{pred}$ is true and 0 otherwise. For two vectors $\boldsymbol{\alpha}$ and $\boldsymbol{\beta}$ we say $\boldsymbol{\alpha} \preceq \boldsymbol{\beta}$ iff $\alpha_i \leq \beta_i$ for all $i = 1, \ldots, n$.

## 2 Background

In sequential prediction with expert advice $\texttt{nature}$ generates elements from an outcome space, $\mathcal{Y}$ while the predictions of the $\texttt{learner}$ and the experts are elements from a prediction space, $\mathcal{D}$ (e.g., we may have $\mathcal{Y} = \{0,1\}$ and $\mathcal{D} = [0,1]$). Given a non-negative loss function $\ell : \mathcal{D} \times \mathcal{Y} \to [0, \infty)$, learning proceeds in trials. On each trial $t = 1, \ldots, T$: 1) the $\texttt{learner}$ receives the expert predictions $\boldsymbol{x}^t \in \mathcal{D}^n$, 2) the $\texttt{learner}$ makes a prediction $\hat{y}^t \in \mathcal{D}$, 3) $\texttt{nature}$ reveals the true label $y^t \in \mathcal{Y}$, and 4) the $\texttt{learner}$ suffers loss $\ell^t := \ell(\hat{y}^t, y^t)$ and expert $i$ suffers loss $\ell_i^t := \ell(x_i^t, y^t)$ for $i = 1, \ldots, n$. Common to the algorithms we consider in this paper is a weight vector, $\boldsymbol{w}^t \in \Delta_n$, where $w_i^t$ can be interpreted as the algorithm's confidence in expert $i$ on trial $t$. The $\texttt{learner}$ uses a prediction function $\texttt{pred} : \Delta_n \times \mathcal{D}^n \to \mathcal{D}$ to generate its prediction $\hat{y}^t = \texttt{pred}(\boldsymbol{w}^t, \boldsymbol{x}^t)$ on trial $t$. A classic example is to predict with the weighted average of the expert predictions, that is, $\texttt{pred}(\boldsymbol{w}^t, \boldsymbol{x}^t) = \boldsymbol{w}^t \cdot \boldsymbol{x}^t$, although for some loss functions improved bounds are obtained with different prediction functions (see e.g., [47]). In this paper we assume $(c, \eta)$-realizability of $\ell$ and $\texttt{pred}$ [4, 18, 45]. That is, there exists constants $c, \eta > 0$ such that for all $\boldsymbol{w} \in \Delta_n$, $\boldsymbol{x} \in \mathcal{D}^n$, and $y \in \mathcal{Y}$, $\ell(\texttt{pred}(\boldsymbol{w}, \boldsymbol{x}), y) \leq -c \ln \sum_{i=1}^n w_i e^{-\eta \ell(x_i, y)}$. This includes $\eta$-exp-concave losses when $\texttt{pred}(\boldsymbol{w}^t, \boldsymbol{x}^t) = \boldsymbol{w}^t \cdot \boldsymbol{x}^t$ and $c = \frac{1}{\eta}$. For simplicity we present regret bound guarantees that assume $(c, \frac{1}{c})$-realizability, that is $c\eta = 1$. This includes the log loss with $c = 1$, and the square loss with $c = \frac{1}{2}$ when $\mathcal{D} = \mathcal{Y} = [0,1]$. The absolute loss is *not* $(c, \eta)$-realizable. Generalizing our bounds to the Hedge setting [13] is straightforward. For any comparison sequence of experts $i_{1:T} = i_1, \ldots, i_T \in [n]$ the regret of the $\texttt{learner}$ with respect to this sequence is defined as

$$\mathcal{R}(i_{1:T}) := \sum_{t=1}^T \ell^t - \sum_{t=1}^T \ell_{i_t}^t \, .$$

We consider and derive algorithms which belong to the family of "exponential weights" (EW) algorithms (see e.g., [25, 35, 47]). After receiving the expert losses the EW algorithm applies the following incremental loss update to the expert weights,

$$\dot{w}_i^t = \frac{w_i^t e^{-\eta \ell_i^t}}{\sum_{j=1}^n w_j^t e^{-\eta \ell_j^t}} \, . \tag{1}$$

**Static setting.** In the static setting the $\texttt{learner}$ competes against a single expert (i.e., $i_1 = \ldots = i_T$). For the static setting the EW algorithm sets $\boldsymbol{w}^{t+1} = \dot{\boldsymbol{w}}^t$ for the next trial, and for $(c, \frac{1}{c})$-realizable losses and prediction functions achieves a static regret bound of $\mathcal{R}(i_{1:T}) \leq c \ln n$.

**Switching.** In the switching (without memory) setting the `learner` competes against a sequence of experts $i_1, \ldots, i_T$ with $k := \sum_{t=1}^{T-1} [i_t \neq i_{t+1}]$ switches. The well-known Fixed-Share algorithm [23] solves the switching problem with the update

$$\boldsymbol{w}^{t+1} = (1-\alpha)\dot{\boldsymbol{w}}^t + \alpha\frac{\mathbf{1}}{n}, \tag{2}$$

by forcing each expert to "share" a fraction of its weight *uniformly* with all experts.[1] The update is parameterized by a "switching" parameter, $\alpha \in [0, 1]$. With an optimally-tuned $\alpha = \frac{k}{T-1}$ the regret with respect to the best sequence of experts with $k$ switches is

$$\mathcal{R}(i_{1:T}) \leq c\left((k+1)\ln n + (T-1)\mathcal{H}\left(\frac{k}{T-1}\right)\right) \leq c\left((k+1)\ln n + k\ln\frac{T-1}{k} + k\right). \tag{3}$$

**Switching with memory.** Freund [12] gave an open problem to improve on the regret bound (3) when the comparison sequence of experts is comprised of a small pool of size $m := |\cup_{t=1}^{T}\{i_t\}| \ll k$. Using counting arguments Freund gave an exponential-time algorithm with the information-theoretic ideal regret bound of $\mathcal{R}(i_{1:T}) \leq c\ln\left(\binom{n}{m}\binom{T-1}{k}m(m-1)^k\right)$, which is upper-bounded by

$$c\left(m\ln n + k\ln\frac{T-1}{k} + (k-m+1)\ln m + k + m\right). \tag{4}$$

The first efficient algorithm solving Freund's problem was presented in the seminal paper [4]. This work introduced the notion of a *mixing scheme*, which is a distribution $\boldsymbol{\gamma}^{t+1}$ with support $\{0, \ldots, t\}$. Given $\boldsymbol{\gamma}^{t+1}$, the algorithm's update on each trial is the *mixture* over all past weight vectors,

$$\boldsymbol{w}^{t+1} = \sum_{q=0}^{t} \gamma_q^{t+1}\dot{\boldsymbol{w}}^q, \tag{5}$$

where $\dot{\boldsymbol{w}}^0 := \frac{1}{n}\mathbf{1}$, and $\gamma_0^1 := 1$. Intuitively, by mixing all "past posteriors" (MPP) the weights of previously well-performing experts can be prevented from vanishing and recover quickly. An efficient mixing scheme requiring $\mathcal{O}(n)$-time per trial is the "*uniform*" mixing scheme given by $\gamma_t^{t+1} = 1-\alpha$ and $\gamma_q^{t+1} = \frac{\alpha}{t}$ for $0 \leq q < t$. A better regret bound was proved with a "*decaying*" mixing scheme, given by

$$\gamma_q^{t+1} = \begin{cases} 1-\alpha & q = t \\ \alpha\frac{1}{(t-q)^\rho}\frac{1}{Z_t} & 0 \leq q < t, \end{cases} \tag{6}$$

where $Z_t = \sum_{q=0}^{t-1}\frac{1}{(t-q)^\rho}$ is a normalizing factor, and $\rho \geq 0$. With a tuning of $\alpha = \frac{k}{T-1}$ and $\rho = 1$ this mixing scheme achieves a regret bound of[2]

$$\mathcal{R}(i_{1:T}) \leq c\left(m\ln n + 2k\ln\frac{T-1}{k} + k\ln(m-1) + k + k\ln\ln(eT)\right). \tag{7}$$

It appeared that to achieve the best regret bounds, the mixing scheme needed to decay towards the past. Unfortunately, computing (6) exactly requires the storage of all past weights, at a cost of $\mathcal{O}(nt)$-time and space per trial. Observe that these schemes set $\gamma_t^{t+1} = 1-\alpha$, where typically $\alpha$ is small, since intuitively switches are assumed to happen infrequently. All updates using such schemes are of the form

$$\boldsymbol{w}^{t+1} = (1-\alpha)\dot{\boldsymbol{w}}^t + \alpha\boldsymbol{v}^t, \tag{8}$$

where $\boldsymbol{v}^t \in \Delta_n$ is a function of the past weights $\dot{\boldsymbol{w}}^0, \ldots, \dot{\boldsymbol{w}}^{t-1}$. We will refer to (8) as the *generalized share update* (see [7]). Fixed-Share is a special case when $\boldsymbol{v}^t = \frac{1}{n}$ for all $t$. This generalized share update features heavily in this paper.

For a decade it remained an open problem to give the MPP update a Bayesian interpretation. This was finally solved in [27] with the use of *partition specialists*. Here on each trial $t$, a specialist

---

[1]Technically in the original Fixed-Share update each expert shares weight to all *other* experts, i.e., $w_i^{t+1} = (1-\alpha)\dot{w}_i^t + \frac{\alpha}{n-1}\sum_{j\neq i}\dot{w}_j^t$. The two updates achieve essentially the same regret bound and are equivalent up to a scaling of $\alpha$.

[2](7) is a simplified upper bound of the bound given in [4, Corollary 9], using $\ln(1+x) \leq x$.

(first introduced in [14]) is either *awake* and predicts in accordance with a prescribed base expert, or is *asleep* and abstains from predicting. For $n$ base experts and finite time horizon $T$ there are $n2^T$ partition specialists. For Freund's problem an assembly of $m$ partition specialists can predict exactly as the comparison sequence of experts. The Bayesian interpretation of the MPP update given in [27, Theorem 2] was simple: to define a mixing scheme $\gamma^{t+1}$ was to induce a prior over this set of partition specialists. The authors of [27] proposed a simple Markov chain prior over the set of partition specialists, giving an efficient $\mathcal{O}(n)$-time per trial algorithm with the regret bound

$$\mathcal{R}(i_{1:T}) \leq c\left[m\ln\frac{n}{m} + m\mathcal{H}\left(\frac{1}{m}\right) + (T-1)\mathcal{H}\left(\frac{k}{T-1}\right) + (m-1)(T-1)\mathcal{H}\left(\frac{k}{(m-1)(T-1)}\right)\right] \quad (9)$$

$$\leq c\left(m\ln n + 2k\ln\frac{T-1}{k} + (k-m+1)\ln m + 2(k+1)\right), \quad (10)$$

which is currently the best known regret bound for Freund's problem. In this work we improve on the bound (9) for tracking experts with memory (Theorem 1). We also show that in fact this Markov prior on partition specialists corresponds to a geometrically-decaying mixing scheme for MPP (Proposition 4). The regret bounds discussed in this paper all rely on optimally tuning one or more parameters, which in practice are usually unknown, and this is true for our regret bound.

Adaptive online learning algorithms with memory have been shown to have better empirical performance than those without memory [15], and to be effective in real-world applications such as intrusion detection systems [39]. While considerable research has been done on switching with memory in online learning (see e.g., [4, 7, 20, 21, 27, 49]), there remain several open problems. Firstly, there remains a gap between the best known regret bound for an efficient algorithm and the information-theoretic ideal bound (4). Present in both bounds (7) and (10) is the factor of 2 in the second term, which does not appear in (4). In [27] this was interpreted as the cost of co-ordination between specialists, essentially one "pays" twice per switch as one specialist falls asleep and another awakens. In this paper we make progress towards closing this gap by avoiding such additional costs the first time each expert is learned by the algorithm. That is, we pay to *remember* but not to *learn*.

Secondly, unless $n$ is very large the current best known bound (9) beats Fixed-Share's bound (3) only when $m \ll k$, but suffers when $m$ is even a moderate fraction of $k$. A natural question is can we improve on Fixed-Share when we relax the assumption that $m \ll k$, and only a few members of a sequence of experts need remembering (consider for instance, $m > k/2$)? In this paper we prove a regret bound that is not only tighter than (9) for all $m$, but for sufficiently large $n$ improves on Fixed-Share for all $m \leq k$. In Figure 1 this behavior is shown for several existing regret bounds and our regret bound.

Our regret bound will hold for two algorithms; one utilizes a weight-sharing update in the sense of (8), and the other utilizes a projection update. Why should we consider projections? Consider for example a large model consisting of many weights, and to update these weights costs time and/or money. Alternatively consider the application of regret-bounded adaptive algorithms in online portfolio selection (see e.g., [32, 44]). Here each "expert" corresponds to a single stock and the weight vector $\boldsymbol{w}^t$ corresponds to a (normalized) portfolio. If $\ell_i^t$ is the negative log return of stock $i$ after day $t$, then the loss function $\ell^t := -\ln\sum_{i=1}^n w_i^t e^{-\ell_i^t}$ is the negative log return of the portfolio. This loss is $(1,1)$-realizable by definition (although there is no prediction function [1]), and the daily price changes in the market naturally induce the "loss update" (1) by updating the portfolio weights. The algorithm's secondary update (projection or weight-sharing) requires the investor to then actively buy/sell to re-balance the portfolio after each trading period, but doing so may incur transaction costs proportional to the amount bought or sold (see e.g., [2, 32]). In Section 3.3 we motivate the use of projections over weight-sharing in this context, proving that projections are strictly more "efficient". Online portfolio selection with transaction costs is an active area of research [9, 30, 32, 33].

## 2.1 Related work

Switching (without memory) in online learning was first introduced in [35] (see also the earlier [34] and independently in the context of universal coding in [43]), and extended with the Fixed-Share algorithm [23]. An extensive literature has built on these works, including but not limited to [1, 4, 7, 8, 16, 17, 21, 22, 24, 27, 29, 38, 42, 49]. Relevant to this work are the results for switching with memory [4, 7, 21, 27, 29, 49]. The first was the seminal work of [4]. The best known result is given in [27], which we improve on. In [49] a reduction of switching with memory to switching without

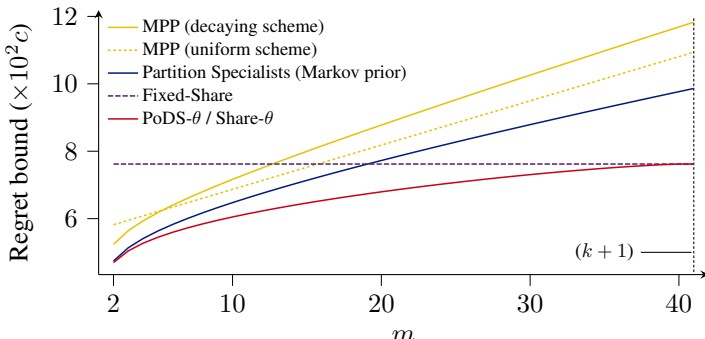

Figure 1: A comparison of the regret bounds discussed in this paper for $m \in [2, k+1]$ with $n = 500000$, $k = 40$, and $T = 4000$. Fixed-Share's bound is constant with respect to $m$. In this case previous "memory" bounds (blue & yellow) are much worse than Fixed-Share for larger values of $m$ while our bound (red) improves on Fixed-Share for all $m \in [2, k]$.

memory is given, although with a slightly worse regret bound than [4]. Related to the experts model is the *bandits* setting, which was addressed in the memory setting in [49].

In [7] a unified analysis of both Fixed-Share and MPP was given in the context of online convex optimization. They observed the generalized share update (8) and slightly improved the bounds of [4]. Adaptive regret [1, 8, 19, 35] has been used to prove regret bounds for switching but unfortunately does not generalize to the memory setting. This paper primarily builds on the work of [4] with a new geometrically-decaying mixing scheme, and on [24] with a new relative entropy projection algorithm. Related to the problem of prediction with expert advice is that of universal coding in information theory (see e.g., [28, 37, 48] for a discussion). Similarly, related to the problem of tracking experts with memory is the problem of universal coding for switching sources with repeating statistics (see e.g., [40, 41, 43] and references therein).

## 3  Projection onto dynamic sets (PoDS)

In this section we give a relative entropy projection-based algorithm for tracking experts with memory. We show how the projection update used by this algorithm is intimately related to the generalized share update (8). In the following subsection we propose a specific update rule for our algorithm for which we improve on the best known regret bound for this problem.

Given a non-empty set $\mathcal{C} \subseteq \Delta_n$ and a point $\boldsymbol{w} \in \mathrm{ri} \, \Delta_n$ we define

$$\mathcal{P}(\boldsymbol{w}; \mathcal{C}) := \arg\min_{\boldsymbol{u} \in \mathcal{C}} D(\boldsymbol{u}, \boldsymbol{w})$$

to be the projection with respect to the relative entropy of $\boldsymbol{w}$ onto $\mathcal{C}$ [6]. Such projections were first introduced for switching (without memory) in online learning in [24], in which after every trial the weight vector $\dot{\boldsymbol{w}}^t$ is projected onto $\mathcal{C} = [\frac{\alpha}{n}, 1]^n \cap \Delta_n$, that is, the simplex with uniform box constraints. For prediction with expert advice this projection algorithm has the regret bound (3) (see [7]). Indeed, we will refer to $\boldsymbol{w}^{t+1} = \mathcal{P}(\dot{\boldsymbol{w}}^t; [\frac{\alpha}{n}, 1]^n \cap \Delta_n)$ as the "projection analogue" of (2). For tracking experts with memory our algorithm will instead project onto a set $\mathcal{C}$ that is updated on each trial, such that each weight does not fall below a certain threshold that is learned for each expert.

Given $\boldsymbol{\beta} \in (0, 1)^n$ such that $\|\boldsymbol{\beta}\|_1 \leq 1$, let

$$\mathcal{C}(\boldsymbol{\beta}) := \{\boldsymbol{x} \in \Delta_n : x_i \geq \beta_i, i = 1, \ldots, n\}$$

be a subset of the simplex which is convex and non-empty. Given $\boldsymbol{w} \in \mathrm{ri} \, \Delta_n$, intuitively $\mathcal{P}(\boldsymbol{w}; \mathcal{C}(\boldsymbol{\beta}))$ is the projection of $\boldsymbol{w}$ onto the simplex with (non-uniform) lower box constraints $\boldsymbol{\beta}$. Relative entropy projection updates for tracking experts with memory were first suggested in [4, Section 5.2]. The authors observed that for any MPP mixing scheme $\boldsymbol{\gamma}^{t+1}$, the update (5) can be replaced with

$$\boldsymbol{w}^{t+1} = \mathcal{P}(\dot{\boldsymbol{w}}^t; \{\boldsymbol{w} \in \Delta_n : \boldsymbol{w} \succeq \gamma_q^{t+1} \dot{\boldsymbol{w}}^q, q = 0, \ldots, t\}), \tag{11}$$

and achieve the same regret bound. We build on this concept in this paper. Observe that for any choice of $\boldsymbol{\gamma}^{t+1}$ the set $\{\boldsymbol{w} \in \Delta_n : \boldsymbol{w} \succeq \gamma_q^{t+1} \dot{\boldsymbol{w}}^q, q = 0, \ldots, t\}$ corresponds to the set $\mathcal{C}(\boldsymbol{\beta})$ where

$$\beta_i = \max_{0 \leq q \leq t} \gamma_q^{t+1} \dot{w}_i^q \qquad (i = 1, \ldots, n) . \tag{12}$$

In this work we give an algorithm to compute the projection $\mathcal{P}(\boldsymbol{w}; \mathcal{C}(\boldsymbol{\beta}))$ exactly for any $\mathcal{C}(\boldsymbol{\beta})$ in $\mathcal{O}(n)$ time (Algorithm 3). With this algorithm and the mapping (12), one immediately obtains the projection analogue of MPP for any mixing scheme $\boldsymbol{\gamma}^{t+1}$ at essentially no additional computational cost. We point out however that for arbitrary mixing schemes computing $\boldsymbol{\beta}$ from (12) takes $\mathcal{O}(nt)$-time on trial $t$, improving only when some structure of the scheme can be exploited. We therefore propose the following method of **P**rojection **o**nto **D**ynamic **S**ets ("PoDS") for tracking experts with memory *efficiently*.

Just as (8) generalizes the Fixed-Share update (2), we propose PoDS as the analogous generalization of the update $\boldsymbol{w}^{t+1} = \mathcal{P}(\dot{\boldsymbol{w}}^t; \mathcal{C}(\alpha\frac{1}{n}))$ (the projection analogue of Fixed-Share). PoDS maintains a vector $\boldsymbol{\beta}^t \in \Delta_n^\alpha$, and on each trial updates the weights by setting $\boldsymbol{w}^{t+1} = \mathcal{P}(\dot{\boldsymbol{w}}^t; \mathcal{C}(\boldsymbol{\beta}^t))$. Intuitively PoDS is the projection analogue of (8) with $\boldsymbol{\beta}^t$ corresponding simply to $\alpha\boldsymbol{v}^t$. In some cases $\boldsymbol{\beta}^t = \alpha\boldsymbol{v}^t$ for all $t$ (e.g., for Fixed-Share), but in general equality may not hold since $\boldsymbol{\beta}^t$ and $\boldsymbol{v}^t$ can be functions of past weights, which may differ for weight-sharing and projection algorithms. Recall that (8) encapsulates all MPP mixing schemes that set $\gamma_t^{t+1} = 1 - \alpha$. PoDS implicitly captures the projection analogue of all such mixing schemes. This simple formulation of PoDS allows us to define new updates, which will correspond to new mixing schemes. In the following section we give a simple update for PoDS and improve on the best known regret bound. In Section 3.2 we discuss Algorithm 3 and the efficient computation of the projection $\mathcal{P}(\boldsymbol{w}; \mathcal{C}(\boldsymbol{\beta}))$.

## 3.1 A simple update rule for PoDS

We now suggest a simple update rule for $\boldsymbol{\beta}^t$ in PoDS for tracking experts with memory. The regret bound for this algorithm is given in Theorem 1 (see Figure 1). We first set $\boldsymbol{\beta}^1 = \alpha\frac{1}{n}$ to be uniform, and with a parameter $0 \leq \theta \leq 1$ update $\boldsymbol{\beta}^t$ on subsequent trials by setting

$$\boldsymbol{\beta}^{t+1} = (1 - \theta)\boldsymbol{\beta}^t + \theta\alpha\dot{\boldsymbol{w}}^t . \tag{13}$$

We refer to PoDS with this update as PoDS-$\theta$. Intuitively the constraint vector $\boldsymbol{\beta}^t$ is updated in (13) by mixing in a small amount of the current weight vector, $\dot{\boldsymbol{w}}^t$, scaled such that $\|\boldsymbol{\beta}^{t+1}\|_1 = \alpha$. If expert $i$ predicted well in the past, then its constraint $\beta_i^t$ will be relatively large, preventing the weight from vanishing even if that expert suffers large losses locally. Using Algorithm 3 in its projection step, PoDS-$\theta$ has $\mathcal{O}(n)$ per-trial time complexity.

As discussed, the vector $\boldsymbol{\beta}^t$ of PoDS is conceptually equivalent to the vector $\alpha\boldsymbol{v}^t$ of the generalized share update (8). If PoDS has a simple update rule such as (13) then it is straightforward to recover the weight-sharing equivalent by simply "pretending" equality holds on all trials. We now do this for PoDS-$\theta$. Clearly we have $\boldsymbol{v}^1 = \frac{1}{n}$, and if $\boldsymbol{\beta}^t = \alpha\boldsymbol{v}^t$ and $\boldsymbol{\beta}^{t+1} = \alpha\boldsymbol{v}^{t+1}$, then $\boldsymbol{v}^{t+1} = \frac{1}{\alpha}\boldsymbol{\beta}^{t+1} = \frac{1}{\alpha}(1-\theta)\boldsymbol{\beta}^t + \theta\dot{\boldsymbol{w}}^t = (1-\theta)\boldsymbol{v}^t + \theta\dot{\boldsymbol{w}}^t$. This then leads to an efficient sharing algorithm, which we call Share-$\theta$. In Section 4 we show this algorithm is in fact a new MPP mixing scheme, which surprisingly corresponds to the previous best known algorithm for this problem. Both PoDS-$\theta$ and Share-$\theta$ use the same parameters ($\alpha$ and $\theta$), differing only in the final update (see Algorithms 1&2).

In the following theorem we give the regret bound which holds for both PoDS-$\theta$ and Share-$\theta$.

**Theorem 1.** *For any comparison sequence $i_1, \ldots, i_T$ containing $k$ switches and consisting of $m$ unique experts from a set of size $n$, if $\alpha = \frac{k}{T-1}$ and $\theta = \frac{k-m+1}{(m-1)(T-2)}$, the regret of both PoDS-$\theta$ and Share-$\theta$ with any prediction function and loss function which are $(c, \frac{1}{c})$-realizable is*

$$\mathcal{R}(i_{1:T}) \leq c\left(m\ln n + (T-1)\mathcal{H}\left(\frac{k}{T-1}\right) + (m-1)(T-2)\mathcal{H}\left(\frac{k-m+1}{(m-1)(T-2)}\right)\right). \tag{14}$$

The regret bound (14) is at least $c((m-1)\ln\frac{T-1}{k} - (k-m+1)\ln\frac{k}{k-m+1})$ tighter than the currently best known bound (9). Thus if $m \ll k$ then the improvement is $\approx cm\ln\frac{T}{k}$, and as $m \to k+1$ then the improvement is $\approx ck\ln\frac{T}{k}$. Additionally note that if $m = k+1$ (i.e., every switch we track a

| **Algorithms 1&2** PoDS-$\theta$ / Share-$\theta$ | **Algorithm 3** $\mathcal{P}(\boldsymbol{w};\mathcal{C}(\boldsymbol{\beta}))$ in $\mathcal{O}(n)$ time |
|---|---|

**Algorithms 1&2** PoDS-$\theta$ / Share-$\theta$

**Input:** $n > 0$, $\eta = \frac{1}{c} > 0$, $\alpha \in [0,1]$, $\theta \in [0,1]$

▷ PoDS-$\theta$
1: **init:** $\boldsymbol{w}^1 \leftarrow \frac{1}{n}$; $\boldsymbol{\beta}^1 \leftarrow \alpha\frac{1}{n}$

▷ Share-$\theta$
1: **init:** $\boldsymbol{w}^1 \leftarrow \frac{1}{n}$; $\boldsymbol{v}^1 \leftarrow \frac{1}{n}$

▷ PoDS-$\theta$ & Share-$\theta$
2: **for** $t \leftarrow 1$ to $T$ **do**
3:    **receive** $\boldsymbol{x}^t \in \mathcal{D}^n$
4:    **predict** $\hat{y}^t = \mathtt{pred}(\boldsymbol{w}^t, \boldsymbol{x}^t)$
5:    **receive** $y^t \in \mathcal{Y}$
6:    **for** $i \leftarrow 1$ to $n$ **do**
7:        $\dot{w}_i^t \leftarrow \frac{w_i^t e^{-\eta \ell_i^t}}{\sum_{j=1}^n w_j^t e^{-\eta \ell_j^t}}$

▷ PoDS-$\theta$
8:    $\boldsymbol{w}^{t+1} \leftarrow \mathcal{P}(\dot{\boldsymbol{w}}^t; \mathcal{C}(\boldsymbol{\beta}^t))$ $\qquad$ (15)
9:    $\boldsymbol{\beta}^{t+1} \leftarrow (1-\theta)\boldsymbol{\beta}^t + \theta\alpha\dot{\boldsymbol{w}}^t$

▷ Share-$\theta$
8:    $\boldsymbol{w}^{t+1} \leftarrow (1-\alpha)\dot{\boldsymbol{w}}^t + \alpha\boldsymbol{v}^t$ $\qquad$ (16)
9:    $\boldsymbol{v}^{t+1} \leftarrow (1-\theta)\boldsymbol{v}^t + \theta\dot{\boldsymbol{w}}^t$ $\qquad$ (17)

**Algorithm 3** $\mathcal{P}(\boldsymbol{w};\mathcal{C}(\boldsymbol{\beta}))$ in $\mathcal{O}(n)$ time

**Input:** $\boldsymbol{w} \in \mathrm{ri}\,\Delta_n$; $\boldsymbol{\beta} \in (0,1)^n$ s.t. $\|\boldsymbol{\beta}\|_1 \leq 1$
**Output:** $\boldsymbol{w}' = \mathcal{P}(\boldsymbol{w};\mathcal{C}(\boldsymbol{\beta}))$

1: **init:** $\mathcal{W} \leftarrow [n]$; $\boldsymbol{r} \leftarrow \boldsymbol{w} \odot \frac{1}{\boldsymbol{\beta}}$; $S_{\boldsymbol{w}} \leftarrow 0$; $S_{\boldsymbol{\beta}} \leftarrow 0$
2: **while** $\mathcal{W} \neq \emptyset$ **do**
3:    $\phi \leftarrow \mathtt{median}(\{r_i : i \in \mathcal{W}\})$
4:    $\mathcal{L} \leftarrow \{i \in \mathcal{W} : r_i < \phi\}$
5:    $L_{\boldsymbol{\beta}} \leftarrow \sum_{i \in \mathcal{L}} \beta_i$; $L_{\boldsymbol{w}} \leftarrow \sum_{i \in \mathcal{L}} w_i$
6:    $\mathcal{M} \leftarrow \{i \in \mathcal{W} : r_i = \phi\}$
7:    $M_{\boldsymbol{\beta}} \leftarrow \sum_{i \in \mathcal{M}} \beta_i$; $M_{\boldsymbol{w}} \leftarrow \sum_{i \in \mathcal{M}} w_i$
8:    $\mathcal{H} \leftarrow \{i \in \mathcal{W} : r_i > \phi\}$
9:    $\lambda \leftarrow \frac{1 - S_{\boldsymbol{\beta}} - L_{\boldsymbol{\beta}}}{1 - S_{\boldsymbol{w}} - L_{\boldsymbol{w}}}$
10:    **if** $\phi\lambda < 1$ **then**
11:        $S_{\boldsymbol{w}} \leftarrow S_{\boldsymbol{w}} + L_{\boldsymbol{w}} + M_{\boldsymbol{w}}$
12:        $S_{\boldsymbol{\beta}} \leftarrow S_{\boldsymbol{\beta}} + L_{\boldsymbol{\beta}} + M_{\boldsymbol{\beta}}$
13:        **if** $\mathcal{H} = \emptyset$ **then**
14:            $\phi \leftarrow \mathtt{min}(\{r_i : r_i > \phi, i \in [n]\})$
15:        $\mathcal{W} \leftarrow \mathcal{H}$
16:    **else**
17:        $\mathcal{W} \leftarrow \mathcal{L}$
18: $\lambda \leftarrow \frac{1 - S_{\boldsymbol{\beta}}}{1 - S_{\boldsymbol{w}}}$
19: $\forall i : 1, \ldots, n : w_i' \leftarrow \begin{cases} \beta_i & r_i < \phi \\ \lambda w_i & r_i \geq \phi \end{cases}$

*new* expert) the optimal tuning of $\theta$ is zero, and PoDS-$\theta$ reduces to setting $\boldsymbol{\beta}^t = \alpha\frac{1}{n}$ on every trial. That is, we recover the projection analogue of Fixed-Share. This is also reflected in the regret bound since (14) reduces to (3). Since $x\mathcal{H}(\frac{y}{x}) \leq y\ln(\frac{x}{y}) + y$, the regret bound (14) is upper-bounded by

$$\mathcal{R}(i_{1:T}) \leq c\left[m\ln n + k\ln\frac{T-1}{k} + (k-m+1)\ln\frac{T-2}{k-m+1} + (k-m+1)\ln(m-1) + 2k - m + 1\right].$$

Comparing this to (10), we see that instead of paying $c\ln\frac{T-1}{k}$ *twice* on every switch, we pay $c\ln\frac{T-1}{k}$ once per switch and $c\ln\frac{T-2}{k-m+1}$ for every switch we *remember* an old expert ($k - m + 1$ times). Unlike previous results for tracking experts with memory, PoDS-$\theta$ and its regret bound (14) smoothly interpolate between the two switching settings. That is, it is capable of exploiting memory when necessary and on the other hand does not suffer when memory is not necessary (see Figure 1).

### 3.2 Computing $\mathcal{P}(\boldsymbol{w};\mathcal{C}(\boldsymbol{\beta}))$

Before we consider PoDS-$\theta$ and Share-$\theta$ further, we briefly discuss the computation of the projection $\mathcal{P}(\boldsymbol{w};\mathcal{C}(\boldsymbol{\beta}))$. In [24] the authors showed that computing relative entropy projection onto the simplex with *uniform* box constraints is non-trivial, but gave an algorithm to compute it in $\mathcal{O}(n)$ time. We give a generalization of their algorithm to compute $\mathcal{P}(\boldsymbol{w};\mathcal{C}(\boldsymbol{\beta}))$ exactly for any non-empty set $\mathcal{C}(\boldsymbol{\beta})$ in $\mathcal{O}(n)$ time. As far as we are aware our method to compute exact relative entropy projection onto the simplex with non-uniform (lower) box constraints in linear time is the first, and may be of independent interest (see e.g., [31]).

We first develop intuition by sketching out the form that $\mathcal{P}(\boldsymbol{w};\mathcal{C}(\boldsymbol{\beta}))$ must take, and then describe how Algorithm 3 computes this projection efficiently. This is stated formally in Theorem 2, the proof of which is given in Appendix B. Firstly consider the case that $\boldsymbol{w} \in \mathcal{C}(\boldsymbol{\beta})$, then trivially $\mathcal{P}(\boldsymbol{w};\mathcal{C}(\boldsymbol{\beta})) = \boldsymbol{w}$, due to the non-negativity of $D(\boldsymbol{u}, \boldsymbol{w})$ and the fact that $D(\boldsymbol{u}, \boldsymbol{w}) = 0$ iff $\boldsymbol{u} = \boldsymbol{w}$ [6]. For the case that $\boldsymbol{w} \notin \mathcal{C}(\boldsymbol{\beta})$, this implies that the set $\{i \in [n] : w_i < \beta_i\}$ is non-empty. For each index $i$ in this set the projection of $\boldsymbol{w}$ onto $\mathcal{C}(\boldsymbol{\beta})$ must set the component $w_i$ to its corresponding constraint value $\beta_i$. The remaining components are then normalized, such that $\sum_{i=1}^n w_i = 1$. However, doing so may cause one (or more) of these components $w_j$ to drop below its constraint $\beta_j$. In Appendix B we prove that the projection algorithm must find the set of components $\Psi$ of least cardinality to set to

their constraint values such that when the remaining components are normalized, no component lies below its constraint, and that this can be done in linear time.

Consider the following inefficient approach to finding $\Psi$. Given $\boldsymbol{w}$ and $\mathcal{C}(\boldsymbol{\beta})$, let $\boldsymbol{r} = \boldsymbol{w} \odot \frac{1}{\boldsymbol{\beta}}$ be a "ratio vector". First sort $\boldsymbol{r}$ in ascending order, and then sort $\boldsymbol{w}$ and $\boldsymbol{\beta}$ according to the ordering of $\boldsymbol{r}$. If $r_1 \geq 1$ then $\Psi = \emptyset$ and we are done ($\Rightarrow \boldsymbol{w} \in \mathcal{C}(\boldsymbol{\beta})$). Otherwise for each $a = 1, \dots, n$: 1) let the candidate set $\Psi' = [a]$, 2) let $\boldsymbol{w}' = \boldsymbol{w}$ except for each $i \in \Psi'$ set $w_i' = \beta_i$, 3) re-normalize the remaining components of $\boldsymbol{w}'$, and 4) let $\boldsymbol{r}' = \boldsymbol{w}' \odot \frac{1}{\boldsymbol{\beta}}$. The set $\Psi$ is then the candidate set $\Psi'$ of least cardinality such that $\boldsymbol{r}' \succeq \mathbf{1}$. This approach requires sorting $\boldsymbol{r}$ and therefore even an efficient implementation takes $\mathcal{O}(n \log n)$ time. Algorithm 3 finds $\Psi$ without having to sort $\boldsymbol{r}$. It instead specifies $\Psi$ uniquely with a threshold, $\phi$, such that $\Psi = \{i : r_i < \phi\}$. Algorithm 3 finds $\phi$ through repeatedly bisecting the set $\mathcal{W} = [n]$ by finding the median of the set $\{r_i : i \in \mathcal{W}\}$ (which can be done in $\mathcal{O}(|\mathcal{W}|)$ time [3]), and efficiently testing this value as the candidate threshold on each iteration. The smallest valid threshold then specifies the set $\Psi$. The following theorem states the time complexity of the algorithm and the form of the projection, which is used in the proof of Theorem 1 (the proof of Theorem 2 is in Appendix B, where we also give a more detailed description of Algorithm 3).

**Theorem 2.** *For any $\boldsymbol{\beta} \in (0, 1)^n$ such that $\|\boldsymbol{\beta}\|_1 \leq 1$, and for any $\boldsymbol{w} \in \mathrm{ri}\, \Delta_n$, let $\boldsymbol{p} = \mathcal{P}(\boldsymbol{w}; \mathcal{C}(\boldsymbol{\beta}))$, where $\mathcal{C}(\boldsymbol{\beta}) = \{\boldsymbol{x} \in \Delta_n : x_i \geq \beta_i, i = 1, \dots, n\}$. Then $\boldsymbol{p}$ is such that for all $i = 1, \dots, n$,*

$$p_i = \max \left\{ \beta_i; \frac{1 - \sum_{j \in \Psi} \beta_j}{1 - \sum_{j \in \Psi} w_j} w_i \right\}, \tag{18}$$

*where $\Psi := \{i \in [n] : p_i = \beta_i\}$. Furthermore, Algorithm 3 computes $\boldsymbol{p}$ in $\mathcal{O}(n)$ time.*

### 3.3 Projection vs. sharing in online learning

We now briefly consider the two types of updates discussed in this paper (projection and weight-sharing) when updating weights may incur costs. Recall the motivating example introduced in Section 2 was in online portfolio selection with transaction costs. It is straightforward to show that in this model transaction costs are proportional to the $\ell_1$-norm of the difference in the weight vectors before and after re-balancing. In Theorem 3 we give a result which in this context guarantees the "cost" of projecting is less than that of weight-sharing.

To compare the update of PoDS and the generalized share update (8), we must consider for a set of weights $\dot{\boldsymbol{w}}^t$, the point $\mathcal{P}(\dot{\boldsymbol{w}}^t; \mathcal{C}(\boldsymbol{\beta}^t))$ and the point $(1 - \alpha)\dot{\boldsymbol{w}}^t + \alpha \boldsymbol{v}^t$. However these points depend on $\boldsymbol{\beta}^t$ and $\boldsymbol{v}^t$ respectively, which may themselves be functions of previous weight vectors $\dot{\boldsymbol{w}}^1, \dots, \dot{\boldsymbol{w}}^{t-1}$, which as discussed are generally not the same for each of the two algorithms. To compare the two updates equally we therefore assume that the current weights are the same (i.e., they must both update the same weights $\dot{\boldsymbol{w}}^t$), and additionally that $\boldsymbol{\beta}^t = \alpha \boldsymbol{v}^t$. The following theorem states that under mild conditions, PoDS is strictly less "expensive" than its weight-sharing counterpart.

**Theorem 3.** *Let $0 < \alpha < 1$. Then for any $\boldsymbol{v} \in \mathrm{ri}\, \Delta_n$, let $\boldsymbol{\beta} = \alpha \boldsymbol{v}$, and for any $\boldsymbol{w} \in \mathrm{ri}\, \Delta_n$ such that $\boldsymbol{w} \neq \boldsymbol{v}$, let $S(\boldsymbol{w}, \boldsymbol{v}) := (1 - \alpha)\boldsymbol{w} + \alpha \boldsymbol{v}$. Then,*

$$\|\mathcal{P}(\boldsymbol{w}; \mathcal{C}(\boldsymbol{\beta})) - \boldsymbol{w}\|_1 < \|S(\boldsymbol{w}, \boldsymbol{v}) - \boldsymbol{w}\|_1 .$$

Thus if one has to pay to update weights, projection is the economical choice.

## 4 A geometrically-decaying mixing scheme for MPP

In this section we look more closely at Share-$\theta$. We show that it is in fact a new type of *decaying* MPP mixing scheme which corresponds to the partition specialist algorithm with Markov prior.

Recall that the previous best known mixing scheme for MPP is the decaying scheme (6). Observe that in (6) the decay (with the "distance" to the current trial $t$) follows a power-law, and that computing (6) exactly takes $\mathcal{O}(nt)$ time per trial. We now derive an explicit MPP mixing scheme from the updates (16) and (17) of Share-$\theta$. Observe that if we define $\dot{\boldsymbol{w}}^0 := \frac{1}{n}$, then an iterative expansion of (17) on any trial $t$ gives $\boldsymbol{v}^t = \sum_{q=0}^{t-1} \theta^{[q \neq 0]}(1 - \theta)^{t-q-1} \dot{\boldsymbol{w}}^q$, from which (16) implies

$\boldsymbol{w}^{t+1} = (1-\alpha)\dot{\boldsymbol{w}}^t + \alpha\boldsymbol{v}^t = \sum_{q=0}^{t} \gamma_q^{t+1}\dot{\boldsymbol{w}}^q$, where

$$\gamma_q^{t+1} = \begin{cases} 1-\alpha & q = t \\ \theta(1-\theta)^{t-q-1}\alpha & 1 \leq q < t \\ (1-\theta)^{t-1}\alpha & q = 0 \,. \end{cases} \tag{19}$$

Note that (19) is a valid mixing scheme since for all $t$, $\sum_{q=0}^{t} \gamma_q^{t+1} = 1$. The Share-$\theta$ update is therefore a new kind of decaying mixing scheme. In this new scheme the decay is *geometric*, and can therefore be computed efficiently, requiring only $\mathcal{O}(n)$ time and space per trial as we have shown. Furthermore MPP with this scheme has the improved regret bound (14).

Another interesting difference between the decaying schemes (19) and (6) is that when $\theta$ is small then (19) keeps $\gamma_0^{t+1}$ relatively large initially and slowly decays this value as $t$ increases. Intuitively by heavily weighting the initial uniform vector $\dot{\boldsymbol{w}}^0$ on each trial early on, the algorithm can "pick up" the weights of new experts easily. Finally as in the case of PoDS-$\theta$, if $m = k + 1$, then with the optimal tuning of $\theta = 0$, this update reduces to the Fixed-Share update (2).

**Revisiting partition specialists.**   We now turn our attention to the previous best known result for tracking experts with memory (the partition specialists algorithm with a Markov prior [27]).

For sleep/wake patterns $(\chi_1 \ldots \chi_T)$ the Markov prior is a Markov chain on states $\{w, s\}$, defined by the initial distribution $\boldsymbol{\pi} = (\pi_w, \pi_s)$ and transition probabilities $P_{ij} := P(\chi_{t+1} = j | \chi_t = i)$ for $i, j \in \{w, s\}$. The algorithm with these inputs efficiently collapses one weight per specialist down to two weights per expert. These two weight vectors, which we denote $\boldsymbol{a}_t$ and $\boldsymbol{s}_t$, represent the total weight of all awake and sleeping specialists associated with each expert, respectively. Note that the vectors $\boldsymbol{a}_t$ and $\boldsymbol{s}_t$ are not in $\Delta_n$, but rather the vector $(\boldsymbol{a}_t, \boldsymbol{s}_t) \in \Delta_{2n}$ and the "awake vector" $\boldsymbol{a}_t$ gets normalized upon prediction. The weights are initialized by setting $\boldsymbol{a}_1 = \pi_w \frac{\mathbf{1}}{n}$, and $\boldsymbol{s}_1 = \pi_s \frac{\mathbf{1}}{n}$. The update[3] of these weights after receiving the true label $y^t$ is given by

$$a_i^{t+1} = P_{ww} \frac{a_i^t e^{-\eta\ell_i^t}(\sum_{j=1}^{n} a_j^t)}{\sum_{j=1}^{n} a_j^t e^{-\eta\ell_j^t}} + P_{sw}s_i^t, \qquad \text{and} \qquad s_i^{t+1} = P_{ws} \frac{a_i^t e^{-\eta\ell_i^t}(\sum_{j=1}^{n} a_j^t)}{\sum_{j=1}^{n} a_j^t e^{-\eta\ell_j^t}} + P_{ss}s_i^t$$

for $i = 1, \ldots, n$. Recall that the authors of [27] proved that an MPP mixing scheme implicitly induces a prior over partition specialists. The following states that the Markov prior is induced by (19).

**Proposition 4.** *Let $0 < \alpha < 1$, and $0 < \theta < 1$. Then the partition specialists algorithm with Markov prior parameterized with $P_{sw} = \theta$, $P_{ws} = \alpha$, $\pi_w = \frac{\theta}{\alpha+\theta}$, and $\pi_s = \frac{\alpha}{\alpha+\theta}$ is equivalent to Share-$\theta$ parameterized with $\alpha$ and $\theta$.*

The proof (given in Appendix E) amounts to showing for all $t$ that $\frac{\boldsymbol{a}_t}{\pi_w} = \boldsymbol{w}^t$ and $\frac{\boldsymbol{s}_t}{\pi_s} = \boldsymbol{v}^t$. The Markov prior on partition specialists therefore corresponds to a geometrically-decaying MPP mixing scheme! Note however that we have proved a tighter regret bound for this algorithm in Theorem 1.

## 5   Discussion

We gave an efficient projection-based algorithm for tracking experts with memory for which we proved the best known regret bound. We also gave an algorithm to compute relative entropy projection onto the simplex with non-uniform (lower) box constraints exactly in $\mathcal{O}(n)$ time, which may be of independent interest. We showed that the weight-sharing equivalent of our projection-based algorithm is in fact a geometrically-decaying mixing scheme for *Mixing Past Posteriors* [4]. Furthermore we showed that this mixing scheme corresponds exactly to the previous best known result (the partition specialists algorithm with Markov prior [27]), and we therefore improved on their bound. We proved a guarantee favoring projection updates over weight-sharing when updating weights may incur costs, such as in online portfolio selection with proportional transaction costs. We are currently applying PoDS-$\theta$ to this problem, primarily extending the work of [44] in the sense of incorporating both the assumption of "memory" and transaction costs.

In this work we focused on proving good regret bounds, which naturally required optimally-tuned parameters. A limitation of our work is that in practice the optimal parameters are unknown. This is

---

[3]In [27] the algorithm is presented in terms of probabilities with the log loss. Here we give the update generalized to $(c, \eta)$-realizable losses.

a common issue in online learning, and one may employ standard techniques to address this such as the "doubling trick", or by using a Bayesian mixture over parameters [46]. For a prominent recent result in this area see [26].

**Ethical considerations.** While the scope of applicability of online learning algorithms is wide, this research in regret-bounded online learning is foundational in nature and we therefore cannot foresee the extent of any societal impacts (positive or negative) this research may have.

## Acknowledgments and Disclosure of Funding

The authors would like to thank the anonymous reviewers for their feedback, insights, and discussion. The authors would also like to thank Dmitry Adamskiy for valuable discussions. This work was supported by the UK Engineering and Physical Sciences Research Council (EPSRC) grant EP/N509577/1.

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
