# A Proof of Theorem 1

*Proof.* We first prove the bound for PoDS-$\theta$, and then prove that Share-$\theta$ has the same bound. We use the relative entropy $D(\boldsymbol{u}^t, \boldsymbol{w}^t)$ as a measure of progress of the algorithm, where $\boldsymbol{u}^t$ is a comparator vector which we take to be a basis vector $\boldsymbol{e}_i$ for some $i \in [n]$ corresponding to the locally best expert $i_t$ in hindsight on trial $t$. Recall that the comparator sequence $i_1, \ldots, i_T$ is partitioned with $k$ *switches* into $k + 1$ *segments*, where a segment is defined as a sequence of trials where the comparator is unchanged, i.e. $i_a = \ldots = i_b$ for some $a < b$.

Recall that `pred` and $\ell$ are assumed to be $(c, \frac{1}{c})$-realizable. That is, for any $\boldsymbol{w}^t \in \Delta_n$, $\boldsymbol{x}^t \in \mathcal{D}^n$, and $y^t \in \mathcal{Y}$, there exists $\eta > 0$ such that

$$\ell(\texttt{pred}(\boldsymbol{w}, \boldsymbol{x}), y) \leq -c \ln \sum_{i=1}^n w_i e^{-\eta \ell(x_i, y)} \tag{20}$$

holds with $c\eta = 1$.

We first establish that

$$\ell^t - \ell^t_{i_t} \leq c \left( D(\boldsymbol{u}^t, \boldsymbol{w}^t) - D(\boldsymbol{u}^t, \dot{\boldsymbol{w}}^t) \right) \tag{21}$$

holds for all $t$. Expanding the relative entropy terms gives

$$
\begin{aligned}
D(\boldsymbol{u}^t, \boldsymbol{w}^t) - D(\boldsymbol{u}^t, \dot{\boldsymbol{w}}^t) &= \sum_{i=1}^n u_i^t \ln \frac{\dot{w}_i^t}{w_i^t} \\
&= \sum_{i=1}^n u_i^t \ln \frac{w_i^t e^{-\eta \ell_i^t}}{w_i^t \sum_{j=1}^n w_j^t e^{-\eta \ell_j^t}} \\
&= -\eta \sum_{i=1}^n u_i^t \ell_i^t - \ln \sum_{j=1}^n w_j^t e^{-\eta \ell_j^t} \\
&\geq -\eta \ell_{i_t}^t + \frac{1}{c} \ell^t,
\end{aligned}
$$

where the inequality follows from (20). Multiplying both sides by $c$ gives (21).

We now find lower bounds, $\delta$, for $D(\boldsymbol{u}^t, \dot{\boldsymbol{w}}^t) - D(\boldsymbol{u}^{t+1}, \boldsymbol{w}^{t+1})$ to give non-negative terms of the form $D(\boldsymbol{u}^t, \dot{\boldsymbol{w}}^t) - D(\boldsymbol{u}^{t+1}, \boldsymbol{w}^{t+1}) - \delta \geq 0$, which we will multiply by $c$ and add to (21) to give a telescoping sum of relative entropy terms. We consider three distinct cases for the different values of $\boldsymbol{u}^t$ over the $T$ trials.

For the first case, we consider when there is no switch immediately after trial $t$ (i.e., $\boldsymbol{u}^t = \boldsymbol{u}^{t+1}$). We use Corollary 7 with $\boldsymbol{u} = \boldsymbol{u}^t$, $\boldsymbol{w} = \dot{\boldsymbol{w}}^t$, and $\boldsymbol{\beta} = \boldsymbol{\beta}^t$. It follows then by definition that $\boldsymbol{p} = \boldsymbol{w}^{t+1}$ and we obtain

$$D(\boldsymbol{u}^t, \dot{\boldsymbol{w}}^t) - D(\boldsymbol{u}^{t+1}, \boldsymbol{w}^{t+1}) \geq \ln(1 - \alpha), \tag{22}$$

which gives a telescoping sum of relative entropy terms within in each segment, paying $c \ln(1/(1-\alpha))$ for every trial where $\boldsymbol{u}^t = \boldsymbol{u}^{t+1}$.

For the two remaining cases, we will consider the segment boundaries, that is, the case when there is a switch and $\boldsymbol{u}^t \neq \boldsymbol{u}^{t+1}$. W.l.o.g let $\boldsymbol{u}^t = \boldsymbol{e}_j$ and let $\boldsymbol{u}^{t+1} = \boldsymbol{e}_k$ for any $j \neq k$ (that is we switch from expert "$j$" to expert "$k$" after trial $t$). We then have the following

$$D(\boldsymbol{u}^t, \dot{\boldsymbol{w}}^t) - D(\boldsymbol{u}^{t+1}, \boldsymbol{w}^{t+1}) = \sum_{i=1}^n u_i^t \ln \frac{u_i^t}{\dot{w}_i^t} - \sum_{i=1}^n u_i^{t+1} \ln \frac{u_i^{t+1}}{w_i^{t+1}} = \ln \frac{1}{\dot{w}_j^t} + \ln w_k^{t+1}, \tag{23}$$

thus we collect a $\ln(1/\dot{w}_j^t)$ term from the *last* trial of the segment of expert $j$ and a $\ln(w_k^{t+1})$ term from the *first* trial of the new segment of expert $k$. We now consider the remaining two cases: when trial $t + 1$ is the first time expert $k$ predicts well, and when trial $t + 1$ is a trial on which we "re-visit" expert $k$.

For the first of these two cases, we consider the first time expert $k$ starts to predict well. We then use (15) and (13) to give

$$\ln w_k^{t+1} \geq \ln \beta_k^t \geq \ln \left( (1-\theta)^{t-1} \beta_k^1 \right) = \ln \left( (1-\theta)^{t-1} \frac{\alpha}{n} \right). \tag{24}$$

Substituting (24) into (23), we therefore pay $-c \ln \left((1-\theta)^{t-1} \frac{\alpha}{n}\right)$ to switch to a new expert for the first time on trial $t + 1$.

Finally for the second of these two cases, we consider when expert $k$ has predicted well before. Let trial $q < t$ denote the *last* trial of expert $k$'s most recent "segment". We then have the following (again using (15) and (13)),

$$\ln w_k^{t+1} \geq \ln \beta_k^t \geq \ln \left((1-\theta)^{t-q-1} \beta_k^{q+1}\right) \geq \ln \left((1-\theta)^{t-q-1} \alpha \theta \dot{w}_k^q\right). \tag{25}$$

By substituting (25) into (23) for each segment boundary, and summing over these boundaries, we therefore pay $-c \ln \left((1-\theta)^{t-q-1} \alpha \theta\right)$ in order to telescope the $\ln \left(\dot{w}_k^q\right)$ term with the $\ln \left(1/\dot{w}_k^q\right)$ term from the end of expert $k$'s most recent segment ending on trial $q$.

Putting these together we thus pay $c \ln \left(1/(1-\alpha)\right)$ for every trial on which we don't switch (from Corollary 7), we pay $c \ln \left(1/(1-\theta)\right)$ for every expert in our pool that *isn't* predicting well or involved in a switch on every trial (i.e., $m - 1$ times, on non-switch trials, and $m - 2$ times on switch trials, from (24) and (25)), and finally when we switch to an expert $k$ before trial $t + 1$ we pay $c \ln \left(n/\alpha\right)$ if it is the first time to track expert $k$ (there are $m - 1$ such trials), and $c \ln \left(1/\alpha\theta\right)$ otherwise (there are $k - m + 1$ such trials).

Summing over all trials, and using $D(\boldsymbol{u}^1, \boldsymbol{w}^1) \leq \ln n$ then gives

$$\sum_{t=1}^T \ell^t - \sum_{t=1}^T \ell_{i_t}^t \leq \sum_{t=1}^T c \left(D(\boldsymbol{u}^t, \boldsymbol{w}^t) - D(\boldsymbol{u}^t, \dot{\boldsymbol{w}}^t) + D(\boldsymbol{u}^t, \dot{\boldsymbol{w}}^t) - D(\boldsymbol{u}^{t+1}, \boldsymbol{w}^{t+1})\right)$$

$$\leq cD(\boldsymbol{u}^1, \boldsymbol{w}^1) + c(T - k - 1) \ln \left(\frac{1}{1-\alpha}\right) + c(m - 1) \ln \left(\frac{n}{\alpha}\right)$$

$$+ c((m-1)(T-1) - k) \ln \left(\frac{1}{1-\theta}\right) + c(k - m + 1) \ln \left(\frac{1}{\alpha\theta}\right)$$

$$\leq cm \ln n + c(T - k - 1) \ln \left(\frac{1}{1-\alpha}\right) + ck \ln \left(\frac{1}{\alpha}\right)$$

$$+ c((m-1)(T-1) - k) \ln \left(\frac{1}{1-\theta}\right) + c(k - m + 1) \ln \left(\frac{1}{\theta}\right). \tag{26}$$

The optimal tuning of $\alpha$ and $\theta$ that minimizes (26) is given by $\alpha = \frac{k}{T-1}$ and $\theta = \frac{k-m+1}{(m-1)(T-2)}$. Substituting these values into (26) gives a bound of

$$cm \ln n + c(T-1)\mathcal{H}\left(\frac{k}{T-1}\right) + c(m-1)(T-2)\mathcal{H}\left(\frac{k-m+1}{(m-1)(T-2)}\right),$$

which completes the proof for PoDS-$\theta$.

We now prove that Share-$\theta$ has the same bound with an almost identical argument as the proof just given for PoDS-$\theta$. Firstly observe that (23) is independent of the algorithm update and therefore holds for both algorithms. Additionally, observe that the proof for PoDS-$\theta$ relies on the inequalities (21), (22), (24), and (25). We now prove that these inequalities hold for Share-$\theta$, and thus the two algorithms share the same bound.

Firstly we observe that inequality (21) holds since both algorithms use the same loss update, and we assume that the prediction function and loss function are $(c, \frac{1}{c})$-realizable.

Secondly, it follows directly from the update (16) that (22) holds for Share-$\theta$ when $\boldsymbol{u}^t = \boldsymbol{u}^{t+1}$, since $\boldsymbol{w}^{t+1} \geq (1-\alpha)\dot{\boldsymbol{w}}^t$ and therefore

$$D(\boldsymbol{u}^t, \dot{\boldsymbol{w}}^t) - D(\boldsymbol{u}^{t+1}, \boldsymbol{w}^{t+1}) = \sum_{i=1}^n u_i^t \ln \frac{w_i^{t+1}}{\dot{w}_i^t} \geq \sum_{i=1}^n u_i^t \ln \frac{(1-\alpha)\dot{w}_i^t}{\dot{w}_i^t} = \ln(1-\alpha).$$

The proof that (24) holds follows directly from the updates (16) and (17) and the fact $\boldsymbol{v}^1 = \frac{1}{n}$. That is, for the first time expert "$k$" appears on trial $t + 1$,

$$\ln w_k^{t+1} \geq \ln \left(\alpha v_k^t\right) \geq \ln \left((1-\theta)^{t-1} \alpha v_k^1\right) = \ln \left((1-\theta)^{t-1} \frac{\alpha}{n}\right).$$

Similarly, the proof that (25) holds follows directly from the updates (16) and (17). That is, when we return to expert "$k$" on trial $t+1$,

$$\ln w_k^{t+1} \geq \ln\left(\alpha v_k^t\right) \geq \ln\left((1-\theta)^{t-q-1}\alpha v_k^{q+1}\right) \geq \ln\left((1-\theta)^{t-q-1}\alpha\theta\dot{w}_k^q\right).$$

Having shown that the inequalities (21), (22), (24), and (25) hold for Share-$\theta$, the remainder of the proof follows exactly as the proof for PoDS-$\theta$. □

## B  Proof of Theorem 2

A note on the proof: The proof of the theorem follows very closely to the proof of Theorem 7 in [24] (including Claims 1, 2, and 3). There the problem is concerned with uniform constraints, whereas we consider non-uniform constraints. In particular Claims 5 and 6 given below are generalizations of Claims 2 and 3 of [24]. The proof of the second statement of Theorem 2 is almost identical to the proof of Theorem 7 in [24]. We first give a sketch of the proof of the two statements of Theorem 2.

For the first statement, recall that $\Psi := \{i \in [n] : p_i = \beta_i\}$ is the set of indexes of components which must be set to their constraint values. To prove the first statement we will show that given $w$ and $\mathcal{C}(\beta)$, each component of the point $\mathcal{P}(w; \mathcal{C}(\beta))$ either takes the value of its lower box constraint, $\beta_i$, or is equal to $w_i$ multiplied by a factor $\lambda$, with

$$\lambda = \frac{1 - \sum_{i \in \Psi} \beta_i}{1 - \sum_{i \in \Psi} w_i}.$$

We then argue that each component $p_i = \max\{\beta_i; \lambda w_i\}$ for $i = 1, \ldots, n$.

For the second statement, we first show that $\Psi$, which uniquely specifies $\mathcal{P}(w; \mathcal{C}(\beta))$, is the set of minimum cardinality such that when all other components are re-normalized, no component lies below its constraint value, and then show that Algorithm 3 finds this set in $\mathcal{O}(n)$ time.

*Proof of the first statement of Theorem 2.*  Recall the first statement of the theorem: that $\mathcal{P}(w; \mathcal{C}(\beta))$ takes the form (18). Given $w$ and the non-empty set $\mathcal{C}(\beta)$, the point $\mathcal{P}(w; \mathcal{C}(\beta))$ is the minimizer of the following convex optimization problem

$$\begin{aligned} \min_{u} \quad & D(u, w) \\ \text{s.t.} \quad & \beta_i - u_i \leq 0, \quad i = 1, \ldots, n \\ & \mathbf{1} \cdot u - 1 = 0 \end{aligned} \tag{27}$$

Since $D(u, w)$ is convex in its first argument, and $\mathcal{C}(\beta)$ is a convex set, then (27) has a unique minimizer, which we denote by $p$.

Constructing the Lagrangian of (27) with Lagrange multipliers $\xi \succeq 0, \nu \in \mathbb{R}$,

$$\mathcal{L}(u, \xi, \nu) = \sum_{i=1}^{n} u_i \ln \frac{u_i}{w_i} + \xi^\top(\beta - u) + \nu(\mathbf{1} \cdot u - 1),$$

and setting $\nabla_u \mathcal{L}(u, \xi, \nu) = \mathbf{0}$ gives for $i = 1, \ldots, n$,

$$\frac{\partial \mathcal{L}}{\partial u_i} = \ln \frac{u_i}{w_i} + 1 - \xi_i + \nu = 0.$$

This then gives for $i = 1, \ldots, n$,

$$p_i = w_i e^{\xi_i - 1 - \nu}.$$

Since $D(u, w)$ is convex in its first argument, and (27) has only linear constraints then strong duality holds and we may exploit the complementary slackness Karush-Kuhn-Tucker necessary condition of the optimal solution (see e.g., [5, Chapter 5]). That is, $\xi_i(\beta_i - p_i) = 0$ for all $i = 1, \ldots, n$. Therefore for any $i$ such that $p_i > \beta_i$, the corresponding Lagrange multiplier is zero, and we have

$$p_i = w_i e^{-1 - \nu}.$$

Recall $\Psi = \{i : p_i = \beta_i\}$, we then have

$$1 = \sum_{i=1}^{n} p_i = \sum_{i \in \Psi} p_i + \sum_{i \in [n] \setminus \Psi} p_i = \sum_{i \in \Psi} \beta_i + \sum_{i \in [n] \setminus \Psi} w_i e^{-1 - \nu}.$$

Re-arranging gives

$$e^{-1-\nu} = \frac{1 - \sum_{i \in \Psi} \beta_i}{\sum_{i \in [n] \setminus \Psi} w_i} = \frac{1 - \sum_{i \in \Psi} \beta_i}{1 - \sum_{i \in \Psi} w_i}.$$

Therefore for each index $i \in [n]$, either $i$ is in $\Psi$ which implies $p_i = \beta_i$, or $i \notin \Psi$ and therefore $p_i = \lambda w_i$, where

$$\lambda = \frac{1 - \sum_{j \in \Psi} \beta_j}{1 - \sum_{j \in \Psi} w_j}.$$

We now establish that $p_i = \max\{\beta_i; \lambda w_i\}$ for all $i = 1, \ldots, n$. Observe that if $i \in \Psi$, then $p_i = w_i e^{\xi_i - 1 - \nu} = \beta_i$, and since the Lagrange multiplier $\xi_i \geq 0$ then $p_i \geq w_i e^{-1-\nu} = \lambda w_i$.

For $i \notin \Psi$, then this implies $p_i = \lambda w_i > \beta_i$, since if $p_i = \beta_i$ then $i \in \Psi$, and if $p_i < \beta_i$ then we have a contradiction since $p$ is not a feasible solution to (27). We therefore conclude that $p$ is such that for all $i = 1, \ldots, n$,

$$p_i = \max\left\{\beta_i; \frac{1 - \sum_{j \in \Psi} \beta_j}{1 - \sum_{j \in \Psi} w_j} w_i\right\},$$

which completes the proof of the first statement of the Theorem. $\qquad\square$

The proof of the second statement of Theorem 2 will rely on the following two claims.

**Claim 5.** *Given $w$ and $\beta$, let $r := w \odot \frac{1}{\beta}$. Without loss of generality, for $i < j$ assume $r_i \leq r_j$. Let $\lambda = \frac{1 - \sum_{i \in \Psi} \beta_i}{1 - \sum_{i \in \Psi} w_i}$, then*

$$p = \left(\beta_1, \ldots, \beta_{|\Psi|}, \lambda w_{|\Psi|+1}, \ldots, \lambda w_n\right). \tag{28}$$

*Proof.* In the proof of the first statement of Theorem 2 we established that $p$ is a permutation of (28), that is, either $p_i = \beta_i$ or $p_i = \lambda w_i$ for $i = 1, \ldots, n$. We also established that $p_i = \max\{\beta_i; \lambda w_i\}$ for $i = 1, \ldots, n$.

Suppose $p$ is not in the form of (28). Then there exists $a < b$ such that $p_a = \lambda w_a$ and $p_b = \beta_b$ (that is, $b \in \Psi$ and $a \notin \Psi$).

If $p_a = \lambda w_a$ then by the first statement of Theorem 2 we have $\lambda w_a > \beta_a$. However since $r_a \leq r_b$, and $\lambda > 0$, this implies $\frac{\lambda w_a}{\beta_a} \leq \frac{\lambda w_b}{\beta_b}$. We then have $1 < \frac{\lambda w_a}{\beta_a} \leq \frac{\lambda w_b}{\beta_b}$, which implies $\lambda w_b > \beta_b$. However we necessarily assumed that $p_b = \beta_b$. This violates the first statement of Theorem 2 that $p_b = \max\{\lambda w_b, \beta_b\}$, and thus contradicts our assumption that $p$ is the minimizer of (27). Hence our supposition that $p$ is not in the form of (28) is false. $\qquad\square$

**Claim 6.** *Let $\Psi' = \{1, \ldots, k\}$, and $\Psi'' = \{1, \ldots, k+1\}$, and let $\lambda' = \frac{1 - \sum_{i \in \Psi'} \beta_i}{1 - \sum_{i \in \Psi'} w_i}$, and $\lambda'' = \frac{1 - \sum_{i \in \Psi''} \beta_i}{1 - \sum_{i \in \Psi''} w_i}$. Then let*

$$u' = \left(\overbrace{\beta_1, \ldots, \beta_{|\Psi'|}}^{k}, \lambda' w_{|\Psi'|+1}, \ldots, \lambda' w_n\right),$$

*and*

$$u'' = \left(\overbrace{\beta_1, \ldots, \beta_{|\Psi''|}}^{k+1}, \lambda'' w_{|\Psi''|+1}, \ldots, \lambda'' w_n\right),$$

*then $D(u', w) \leq D(u'', w)$.*

*Proof.* Consider the following convex optimization problem for some $w \in \mathrm{ri}\, \Delta_n$,

$$
\begin{aligned}
\min_{u} \quad & D(u, w) \\
\text{s.t.} \quad & \beta_i - u_i = 0, \quad i = 1, \ldots, k \\
& \mathbf{1} \cdot u - 1 = 0 \quad .
\end{aligned}
\tag{29}
$$

The point $u'$ is the unique minimizer of (29), while $u''$ clearly also satisfies the constraints of (29) and is therefore a feasible solution. This implies that $D(u', w) \leq D(u'', w)$. $\qquad\square$

*Proof of the second statement of Theorem 2.* Recall the second statement of the theorem: that Algorithm 3 computes $\mathcal{P}(\boldsymbol{w}; \mathcal{C}(\boldsymbol{\beta}))$ in linear time. We prove this statement by first showing that the set $\Psi$ corresponding to this projection is the set of components of minimal cardinality to set to their constraint values such that when the other components are normalized, no component lies below its constraint value. We then prove that Algorithm 3 computes the projection by finding this set in linear time.

In the proof of the first statement of the theorem we proved that $\boldsymbol{p}$ has the form (18). Thus $\boldsymbol{p}$ is uniquely specified by the set $\Psi = \{i \in [n] : p_i = \beta_i\} \subseteq \{1, \ldots, n\}$. There are therefore $2^n$ possible solutions. Claim 5 proves that the magnitude of the ratio of a component and its constraint is smaller for a component to be set to its constraint value than a component to be normalized. That is, if $i \in \Psi$ and $j \notin \Psi$, then $\frac{w_i}{\beta_i} \leq \frac{w_j}{\beta_j}$. This reduces the number of feasible solutions to $n$.

Given these $n$ possible solutions, claim 6 shows that if $\Psi' \subseteq \Psi''$ with corresponding candidate projection vectors $\boldsymbol{u}'$ and $\boldsymbol{u}''$ respectively, then $D(\boldsymbol{u}', \boldsymbol{w}) \leq D(\boldsymbol{u}'', \boldsymbol{w})$. Thus to compute the projection, one must find the set $\Psi$ of minimum cardinality whose corresponding candidate projection vector is in $\mathcal{C}(\boldsymbol{\beta})$.

Observe that this "minimal" set $\Psi$ is specified uniquely by a threshold, $\phi$, such that $\Psi = \{i \in [n] : r_i < \phi\}$, where $r_i = \frac{w_i}{\beta_i}$, for $i = 1, \ldots, n$. Algorithm 3 finds $\Psi$ by finding this threshold. The algorithm initially computes the vector $\boldsymbol{r} = \boldsymbol{w} \odot \frac{1}{\boldsymbol{\beta}}$ and when $\phi$ has been found, the algorithm sets all components of $w_i$ where $r_i < \phi$ to their thresholds $\beta_i$, and normalizes the remaining components.

We now discuss how the algorithm finds $\phi$ in linear time. On each iteration a candidate threshold is examined. These candidate thresholds are determined from an index set $\mathcal{W}$, which is initially set to $\{1, \ldots, n\}$. On each iteration the threshold $\phi$ is chosen as the median of the ratios in the set $\{r_i : i \in \mathcal{W}\}$ (line 3). This can be done in $\mathcal{O}(|\mathcal{W}|)$ time [3]. The approach used is a divide and conquer method, however from a practical perspective this could also be replaced with a randomized median-finding algorithm with average time complexity $\mathcal{O}(|\mathcal{W}|)$ [10]. If $|\mathcal{W}|$ is even, then the algorithm can choose between the $\frac{|\mathcal{W}|}{2}$ and the $\frac{|\mathcal{W}|+1}{2}$ largest element arbitrarily. The set $\mathcal{W}$ is then sorted into two sets, $\mathcal{L}$ and $\mathcal{H}$, where $\mathcal{L} = \{i \in \mathcal{W} : r_i < \phi\}$ and $\mathcal{H} = \{i \in \mathcal{W} : r_i > \phi\}$.

The normalizing constant $\lambda$ is then computed (line 9). If $\lambda\phi < 1$, then by Claims 5 and 6 the true threshold must be larger than the current candidate threshold $\phi$, and must therefore correspond to $r_i$ for an index $i$ contained in $\mathcal{H}$. Otherwise the true threshold must be either equal to the current candidate threshold, or must correspond to $r_i$ for an index $i$ contained in $\mathcal{L}$.

Since $\phi$ was taken to be the median, then the algorithm iterates this procedure, setting $\mathcal{W} = \mathcal{L}$ or $\mathcal{W} = \mathcal{H}$ as appropriate. Additionally, since $\phi$ was taken to be the median, then $\max\{|\mathcal{L}|; |\mathcal{H}|\} \leq \frac{1}{2}|\mathcal{W}|$. When $\mathcal{W} = \emptyset$, then the algorithm has found $\phi$, and the projection is computed.

There are a maximum of $\lceil \log n + 1 \rceil$ iterations of lines 2-17, with the $i^{th}$ iteration taking $\mathcal{O}(\frac{n}{2^i})$ time. The algorithm therefore takes $\mathcal{O}(n)$ time to find $\phi$, and the time complexity of the algorithm is therefore $\mathcal{O}(n)$. $\qquad\square$

## C   Corollary 7

**Corollary 7.** *Let* $0 < \alpha < 1$. *Then for any* $\boldsymbol{u} \in \Delta_n$, $\boldsymbol{w} \in \mathrm{ri}\,\Delta_n$, *and* $\boldsymbol{\beta} \in \mathrm{ri}\,\Delta_n^\alpha$, *let* $\boldsymbol{p} = \mathcal{P}(\boldsymbol{w}; \mathcal{C}(\boldsymbol{\beta}))$. *Then,*

$$D(\boldsymbol{u}, \boldsymbol{w}) - D(\boldsymbol{u}, \boldsymbol{p}) \geq \ln(1 - \alpha). \tag{30}$$

*Proof.* Let $\Psi := \{i \in [n] : p_i = \beta_i\}$. Recall from Theorem 2 that the projected vector $\boldsymbol{p}$ takes the form (18). Expanding the relative entropy terms of (30) then gives the following,

$$
\begin{aligned}
D(\boldsymbol{u}, \boldsymbol{w}) - D(\boldsymbol{u}, \boldsymbol{p}) &= \sum_{i=1}^{n} u_i \ln\left(\frac{p_i}{w_i}\right) \\
&\geq \sum_{i=1}^{n} u_i \ln\left(\frac{\left(1 - \sum_{j \in \Psi} \beta_j\right) w_i}{\left(1 - \sum_{j \in \Psi} w_j\right) w_i}\right) \\
&= \ln\left(\frac{1 - \sum_{j \in \Psi} \beta_j}{1 - \sum_{j \in \Psi} w_j}\right) \\
&\geq \ln(1 - \alpha),
\end{aligned}
$$

where the first inequality follows from the definition of $p_i$ in (18) and the fact that $\max\{a, b\} \geq b$. The second inequality follows from the fact that $\sum_{j \in \Psi} w_j \geq 0$ and $\sum_{j \in \Psi} \beta_j \leq \alpha$. $\qquad \square$

## D   Proof of Theorem 3

Before proving Theorem 3, we introduce some additional notation. Let $\boldsymbol{p} := \mathcal{P}(\boldsymbol{w}; \mathcal{C}(\boldsymbol{\beta}))$, and for brevity let $\boldsymbol{w}' := (1 - \alpha)\boldsymbol{w} + \alpha\boldsymbol{v}$. We then define the following sets,

$$
\begin{aligned}
\mathcal{P}_{inc} &:= \{i \in [n] : p_i > w_i\}, \quad \mathcal{P}_{dec} := \{i \in [n] : p_i \leq w_i\}, \\
\mathcal{S}_{inc} &:= \{i \in [n] : w_i' > w_i\}, \quad \mathcal{S}_{dec} := \{i \in [n] : w_i' \leq w_i\}.
\end{aligned}
$$

The subscripts $inc$ and $dec$ correspond to the relative change in the weights before and after the corresponding update - whether they *increase* or *decrease*, respectively.

We first require the following corollary, which follows naturally from Theorem 2.

**Corollary 8.** *If* $i \in \mathcal{P}_{inc}$ *then* $p_i = \beta_i$.

*Proof.* Recall that Theorem 2 states that $\boldsymbol{p}$ is such that for $i = 1, \ldots, n$,

$$p_i = \max\{\beta_i; \lambda w_i\},$$

where $\lambda = \frac{1 - \sum_{j \in \Psi} \beta_j}{1 - \sum_{j \in \Psi} w_j}$ is a normalizing constant. We first establish that $\lambda \leq 1$. Suppose $\lambda > 1$, then this implies $\sum_{i \in \Psi} w_i > \sum_{i \in \Psi} \beta_i$. In this case there must exist $i \in \Psi$ such that $w_i > \beta_i$. However if $\lambda > 1$ then $\lambda w_i > w_i > \beta_i$, but since $i \in \Psi$ then $p_i = \beta_i$, which must be greater than $\lambda w_i$ by Theorem 2. This leads to a contradiction and thus our supposition that $\lambda > 1$ is false.

The form of $\boldsymbol{p}$ implies that $i \in \mathcal{P}_{inc}$ iff $w_i < \beta_i$, since if $w_i \geq \beta_i$ then this implies that either $p_i = \beta_i \leq w_i$ or $p_i = \lambda w_i \leq w_i$, and in both of these cases $i$ must be in $\mathcal{P}_{dec}$. It then follows that if $i \in \mathcal{P}_{inc}$ then $p_i = \beta_i$ since otherwise $p_i = \lambda w_i \leq w_i < \beta_i$ which is a contradiction. $\qquad \square$

Recall that is it assumed that $\boldsymbol{w} \neq \boldsymbol{v}$ and thus the definition of $\boldsymbol{w}'$ implies that $\mathcal{S}_{inc}$ is non-empty. We use this fact in the following two lemmas. The first states that if a weight $w_i$ were to increase after the projection update, then it would always increase after the weight-sharing update.

**Lemma 9.** $\mathcal{P}_{inc} \subseteq \mathcal{S}_{inc}$.

*Proof.* For any $i \in [n]$ we have

$$w_i' - w_i = (1 - \alpha)w_i + \alpha v_i - w_i = \alpha(v_i - w_i),$$

and it follows that $i \in \mathcal{S}_{inc}$ iff $w_i < v_i$. Using Corollary 8 we conclude that if $i \in \mathcal{P}_{inc}$, then $w_i < p_i = \beta_i = \alpha v_i < v_i$ and then $i$ must also be in $\mathcal{S}_{inc}$. $\qquad \square$

**Lemma 10.** $\|\boldsymbol{p} - \boldsymbol{w}\|_1 = 2\sum_{i \in \mathcal{P}_{inc}} (p_i - w_i)$, and $\|\boldsymbol{w}' - \boldsymbol{w}\|_1 = 2\sum_{i \in \mathcal{S}_{inc}} (w_i' - w_i)$.

*Proof.* We prove the first equality by observing that

$$\|\boldsymbol{p} - \boldsymbol{w}\|_1 = \sum_{i=1}^{n} |p_i - w_i| = \sum_{i \in \mathcal{P}_{inc}} (p_i - w_i) + \sum_{i \in \mathcal{P}_{dec}} (w_i - p_i),$$

and since the total weight does not change after an update (i.e., $\sum_{i=1}^{n} p_i = \sum_{i=1}^{n} w_i$), necessarily we have $\sum_{i \in \mathcal{P}_{inc}} (p_i - w_i) = \sum_{i \in \mathcal{P}_{dec}} (w_i - p_i)$. Since $\sum_{i=1}^{n} w_i' = \sum_{i=1}^{n} w_i$, the same argument can be used to prove the second claim. $\qquad\square$

*Proof of Theorem 3.* Using Corollary 8, and the definition of $\boldsymbol{w}'$, we have for $i \in \mathcal{P}_{inc}$,

$$w_i' - w_i = (1-\alpha)w_i + \alpha v_i - w_i = \alpha(v_i - w_i) = \beta_i - \alpha w_i = p_i - \alpha w_i > p_i - w_i, \qquad (31)$$

where the inequality arises from the fact that $\alpha < 1$. Finally combining this inequality with Lemmas 9 and 10 gives

$$\|\boldsymbol{p} - \boldsymbol{w}\|_1 = 2 \sum_{i \in \mathcal{P}_{inc}} (p_i - w_i) \qquad\qquad\text{(Lemma 10)}$$

$$< 2 \sum_{i \in \mathcal{P}_{inc}} (w_i' - w_i) \qquad\qquad\text{(Equation 31)}$$

$$\leq 2 \sum_{i \in \mathcal{S}_{inc}} (w_i' - w_i) \qquad\qquad\text{(Lemma 9)}$$

$$= \|\boldsymbol{w}' - \boldsymbol{w}\|_1 . \qquad\qquad\text{(Lemma 10)}$$

$\qquad\square$

# E    Proof of Proposition 4

*Proof.* It suffices to show that

$$\frac{a_i^t}{\sum_{j=1}^{n} a_j^t} = w_i^t, \qquad (32)$$

and

$$\frac{s_i^t}{\sum_{j=1}^{n} s_j^t} = v_i^t \qquad (33)$$

for all $t$. Since the initial distribution, $\boldsymbol{\pi}$, of the Markov chain prior is taken to be the stationary distribution, the detailed balance equation, $P_{ws}\pi_w = P_{sw}\pi_s$, holds for all trials.

It is therefore straightforward to show that $\sum_{i=1}^{n} a_i^t = \pi_w$ and $\sum_{i=1}^{n} s_i^t = \pi_s$ for all $t$. Letting $\alpha = P_{ws}$, and $\theta = P_{sw}$, we proceed to prove that (32) and (33) hold simultaneously for all $t$ by induction. The case for $t = 1$ is trivial. Then by induction on $t$ for $t \geq 1$,

$$\frac{a_i^{t+1}}{\pi_w} = P_{ww} \frac{a_i^t e^{-\eta \ell_i^t}}{\sum_{j=1}^{n} a_j^t e^{-\eta \ell_j^t}} + \frac{P_{sw}}{\pi_w} s_i^t$$

$$= P_{ww} \frac{a_i^t e^{-\eta \ell_i^t}}{\sum_{j=1}^{n} a_j^t e^{-\eta \ell_j^t}} + \frac{P_{ws}}{\pi_s} s_i^t$$

$$= P_{ww} \dot{w}_i^t + P_{ws} v_i^t \qquad\qquad\text{(induction)}$$

$$= (1-\alpha)\dot{w}_i^t + \alpha v_i^t$$

$$= w_i^{t+1},$$

and similarly

$$
\begin{aligned}
\frac{s_i^{t+1}}{\pi_s} &= \frac{P_{ws}\pi_w}{\pi_s} \frac{a_i^t e^{-\eta \ell_i^t}}{\sum_{j=1}^n a_j^t e^{-\eta \ell_j^t}} + P_{ss} \frac{s_i^t}{\pi_s} \\
&= P_{sw} \frac{a_i^t e^{-\eta \ell_i^t}}{\sum_{j=1}^n a_j^t e^{-\eta \ell_j^t}} + P_{ss} \frac{s_i^t}{\pi_s} \\
&= P_{sw} \dot{w}_i^t + P_{ss} v_i^t \qquad\qquad\qquad\qquad \text{(induction)} \\
&= \theta \dot{w}_i^t + (1 - \theta) v_i^t \\
&= v_i^{t+1} .
\end{aligned}
$$

We therefore conclude by the inductive argument that (32) and (33) hold for all $t \geq 1$. $\qquad\square$