# OpenReview forum: "Improved Regret Bounds for Tracking Experts with Memory"
_NeurIPS.cc/2021/Conference — NeurIPS 2021 Poster_

### Official Review · Reviewer_dt3a · 2021-07-16

**Rating:** 6
**Confidence:** 4

**Summary:**

This paper studies 'prediction with expert advice' in a non-stationary environment where the best expert is switching. The main contribution is an improved regret bound in the 'memory' setting (where a small number of experts is optimal), which is achieved with an efficient algorithm (in fact two variants are proposed). The algorithm is linear in the number of experts and the improvement compared to the best previous efficient algorithm is a factor two in one term of the regret bound, almost matching the best bound for a non-efficient algorithm. Further, the authors introduce a generalized algorithm for computing the projection of w.r.t. the relative entropy in linear time, and give a Bayesian interpretation of the Share-theta algorithm.

**Limitations And Societal Impact:**

As stated also by the authors, one big limitation of the method is that the algorithm needs to be tune using the 'k', 'm' and 'T' parameters. In practice, these are almost always unknown and one would hope to pay a little or no penalty for adaptivity. I would encourage the authors to discuss this point in more detail, and state if there are known techniques to make the algorithm more adaptive (I am aware of the doubling trick, but this seems highly non-trivial to apply here, since doubling usually only avoids the knowledge on the horizon on a sqrt{T} regret bound).

Finally, in my opinion it is a pity that papers (including this one) on online learning rarely include empirical evaluations. The worry is that by only optimizing the theoretical regret bounds, the algorithms are biased to perform well on the worst-case instances, but perhaps unfavorably on average cases or real-world examples.

**Main Review:**

Originality:
The paper introduce several new ideas, including the efficient computation of the relative entropy projection, and the improved regret bounds for PoDS-theta and Share-theta, and the connection between Share-theta and the MPP approach. I liked the detailed introduction and comparison to previous work.

Quality and Clarity:
Generally, the paper is very well written and organized. The improvement on the regret bound however seems rather marginal. It would be good to know if the improvement translates to better empirical performance. Please also clarify if (3) is a lower bound, or simply the best known upper bound. The reduced switching cost of the projection-based approach is an interesting observation. Also the appendix is quite polished and I did not identify any problems with the proofs.

Significance:
This paper makes an important contribution to the online learning literature.

[edit]
Thank you for the detailed responses. I prefer to keep my score.

**Time Spent Reviewing:**

3

---

> ### Author Response · Authors · 2021-08-10
> **Response to Reviewer dt3a**
>
> Thank you for your review and comments. Please find our specific responses to your comments below:
>
> > *The improvement on the regret bound however seems rather marginal. It would be good to know if the improvement translates to better empirical performance.*
>
> As we say in the discussion with reviewer AJRz, we would ideally include simulations but due to the number of theoretical results in the paper we could not incorporate them due to space constraints. We are particularly interested in the application of our projection method to the problem of online portfolio selection with transaction costs (as discussed in e.g., lines 259-264). This research is ongoing.
> Note that simulations comparing the "memory" case to the "non-memory" case exist in [4, Section 6] which we will reference in Section 5 (Discussion).
>
> > *Please also clarify if (3) is a lower bound, or simply the best known upper bound.*
>
> To clarify (3) is the best known upper bound obtained by an inefficient algorithm. To the best of our knowledge there are no known lower bounds for "tracking experts with memory".  This is a good open problem.  We can improve our discussion in the paper by discussing known lower bounds for tracking experts without memory such as [32 (Section 9)] with the "log loss".
>
> > *As stated also by the authors, one big limitation of the method is that the algorithm needs to be tune using the 'k', 'm' and 'T' parameters. In practice, these are almost always unknown and one would hope to pay a little or no penalty for adaptivity. I would encourage the authors to discuss this point in more detail, and state if there are known techniques to make the algorithm more adaptive (I am aware of the doubling trick, but this seems highly non-trivial to apply here, since doubling usually only avoids the knowledge on the horizon on a sqrt{T} regret bound).*
>
> Currently we only have a very short discussion on the "auto-tuning" parameters in lines 324-328, as discussed with reviewer gqjS, we will add a discussion on this in the introduction also.  There is an extensive literature on this in general in online learning and more specifically in the "tracking the best expert.'
>
> A more detailed discussion of this would split into two cases 1) $(c,1/c)$-realizable losses such as the square loss or log loss 2) non $(c,1/c)$-realizable losses such as the absolute loss or the Hedge setting.  The distinction between the two cases is whether the learning rate $\eta$ only depends on the loss function (case 1) or whether the learning rate depends on $T,k,m$.
>
> In the case of $(c,1/c)$-realizable losses $\eta$ is fixed depending on loss function.  The remaining two parameters can $\theta,\alpha$ can be addressed by a variety methods, but many such methods can be reduced to having a prior over the parameters and then "mix" over all parameter settings [39].  This can be done with varying degrees of efficiency at a trade-off between computational complexity and regret.  In fact [26, p.6] claims such a strategy is possible for "tracking with memory" at no asymptotic computational cost and provides a regret bound.  We did not cite that result because it is provided without proof.  However, if we can verify the result we will then cite it.
>
>
>
> > *Finally, in my opinion it is a pity that papers (including this one) on online learning rarely include empirical evaluations. The worry is that by only optimizing the theoretical regret bounds, the algorithms are biased to perform well on the worst-case instances, but perhaps unfavorably on average cases or real-world examples.*
>
> We agree. As discussed previously we are particularly interested in the application of our projection method to the problem of online portfolio selection with transaction costs (as discussed in e.g., lines 259-264). This research is ongoing.
>
> In terms of performance on average cases or real-world examples, we point out the recent open problem of [1] from COLT. With several papers addressing this problem such as [41].
>
> [1] Warmuth M.K., and Koolen W.M.. Open Problem: Shifting Experts on Easy Data. In Proceedings of the 27th Annual Conference on Learning Theory (COLT), 2014.

---

### Official Review · Reviewer_GNro · 2021-07-16

**Rating:** 6
**Confidence:** 2

**Summary:**

This paper studies the tracking experts with memory problem, and proposes a new algorithm with improved regret bound and linear computational complexities wrt the number of experts.

**Ethics Review Area:**

["I don’t know"]

**Limitations And Societal Impact:**

The questions about the limitations of the proposed methods are discussed in the main review.

**Main Review:**

This paper is well-written in general, and the proposed method is novel, which enjoys an improved regret bound with respect to [26], and share the same computational costs. The technique for doing relative entropy projection in linear time is also interesting. I have the following questions:

1.	From the discussion under Theorem 3, it seems that the *order* of regret bound is only improved by constant factors (eq. (10) vs Line 253, $c\ln{\frac{T-1}{k}}$ to $c\ln{\frac{T-2}{k-m+1}}$). Could the authors add more discussions on the significance of the results?
2.	Can the authors compare the computational complexities of the proposed algorithms with existing methods?


**Time Spent Reviewing:**

4

---

> ### Author Response · Authors · 2021-08-10
> **Response to Reviewer GNro**
>
> Thank you for your review and comments. Please find our specific responses to your comments below:
>
> > *From the discussion under Theorem 3, it seems that the order of regret bound is only improved by constant factors (eq. (10) vs Line 253, $c\ln{\frac{T-1}{k}}$ to $c\ln{\frac{T-2}{k-m+1}}$). Could the authors add more discussions on the significance of the results?*
>
> On the regret bound, the significance of the improvement of the bound is primarily in the fact that we do not "pay twice" per switch as in (10) but rather only to "remember" an expert. Thus when $m\ll k$ the improvement is small, however the improvement is most impactful when memory is only required for some experts (see Figure 1).
>
> Additionally Algorithm 3 is an improvement on the state of the art of exact relative entropy projection. The algorithm may be of independent interest to the community.
>
> It is also very surprising to us that the Markov prior used in the partition specialists algorithm in [26] corresponds to the geometrically-decaying mixing scheme of Share-$\theta$.
>
> Finally we believe that Theorem 4 is particularly significant. The proof that projection updates are always more conservative than weight-sharing updates wrt the L1 norm may be of value in different applications. In particular to the problem of portfolio selection as discussed in lines 140-147 (additionally please refer to the discussion with reviewer AJRz on this topic).
>
> > *Can the authors compare the computational complexities of the proposed algorithms with existing methods?*
>
> For PoDS-$\theta$ this is discussed in line 234. Our two algorithms (PoDS-$\theta$ and Share-$\theta$) have $O(n)$ space and time requirements.
> The computational complexity of these algorithms is the same as that the previous best known result [26], and is better than the previous best known mixing scheme (6) from [4], for which exact computation required $O(nt)$ time and space on trial $t$.
>
> In Theorem 1 we prove that surprisingly Algorithm 3 also computes relative entropy projection with non-uniform lower box constraints in $O(n)$ time (requiring $O(n)$ space).

---

### Official Review · Reviewer_AJRz · 2021-07-16

**Rating:** 6
**Confidence:** 4

**Summary:**

This paper considers the problem of tracking the best expert with long-term memory. This corresponds to prediction with (n) expert advice in which the learner's performance is compared to sequences of experts that may switch k times between a few (m) experts only. The authors consider two related algorithms: one introduced by Bousquet and Warmuth (2002), which mixes past posteriors (MPP), and one which is based on projections with respect to the relative entropy. For well-chosen parameters and for a well-chosen newly introduced mixing scheme, they improve for both algorithms the best existing (efficient) upper-bound for this problem by a factor of 2. This closes the gap with respect to the best non-efficient algorithm.

They provide also a linear-time method to compute projections (with respect to the relative entropy) on the simplex with lower box constraints. This allows their algorithms to require only O(n) per-round computation where n is the number of experts.

**Limitations And Societal Impact:**

Yes

**Main Review:**


On the positive side:
- I found the paper very well written and enjoyable to read. Besides the technical contributions, I believe that this paper would provide a good introduction to the problem of tracking the best expert with memory. The proposed algorithms, proofs, and regret bounds are clean and easy to follow.
- The efficient projection algorithm with respect to the relative entropy could be of independent interest for the community.
- The regret upper-bound improves the known no-memory regret bound of fixed share whenever m<k. This shows that memory is always useful which was not the case for previous bounds that required m significantly smaller than k.


On the negative side:
- We could say that the contribution is somewhat limited and that the problem of improving the existing limits by a small constant factor is of interest only to a niche of the community.
- The method requires the beforehand knowledge of unknown parameters: number of switches (k), number of experts used in the memory (m), horizon (T). If I agree that T can be tuned by a doubling trick this is more complicated for k and m. I know that k can be tuned by using sleeping experts techniques. Is it also the case for m?
- I agree that the contribution is mostly theoretical but some simulations could illustrate if there are practical differences between the two schemes (projections vs MPP) and if this factor 2 really makes a difference in real life.


Other comments:
- Is the regret bound also always better than the one of fixed-share when m<n? A figure similar to Figure 1 with n instead of k might be interesting.
- I didn't find the last paragraph before section 2.1 that motivates the use of projection very clear. I understand that there is more detail later but at that point, I had no idea why these examples motivated the use of projection.
- What is the advantage of using (c,1/c)-realizability with respect to exp-concavity? The latter seems more common.

Minor comments:
- State that (3) is for a well-chosen alpha.
- line 70: square loss has c=1/2 only if the prediction and output domains are [0,1]

**Time Spent Reviewing:**

3

---

> ### Author Response · Authors · 2021-08-10
> **Response to Reviewer AJRz**
>
> Thank you for your review and comments. Please find our specific responses to your comments below:
>
> > *The method requires the beforehand knowledge of unknown parameters: number of switches (k), number of experts used in the memory (m), horizon (T). If I agree that T can be tuned by a doubling trick this is more complicated for k and m. I know that k can be tuned by using sleeping experts techniques. Is it also the case for m?*
>
> Currently we only have a very short discussion on the "auto-tuning" parameters in lines 324-328.  There is an extensive literature on this in general in online learning and more specifically in the "tracking the best expert".
>
> A more detailed discussion of this would split into two cases 1) $(c,1/c)$-realizable losses such as the square loss or log loss 2) non $(c,1/c)$-realizable losses such as the absolute loss or the Hedge setting.  The distinction between the two cases is whether the learning rate $\eta$ only depends on the loss function (case 1) or whether the learning rate depends on $T,k,m$.
>
> Case 1) $\eta$ is fixed depending on loss function.  The remaining two parameters $\theta,\alpha$ can be addressed by a variety methods, but many such methods can be reduced to having a prior over the parameters and then "mix" over all parameter settings [39].  This can be done with varying degrees of efficiency at a trade-off between computational complexity and regret.  In fact [26, p.6] claims such a strategy is possible for "tracking with memory" at no asymptotic computational cost and provides a regret bound.  We did not cite that result because it is provided without proof.  However, if we can verify the result we will then cite it.
>
> Case 2) Is not so straightforward as we cannot directly "mix" over $\eta$ and the optimal $\eta$ depends on $T,k,m$.  Note that in this case the "doubling trick" will also not apply directly for tracking in general. We explain this intuitively in terms of a generalization of the Hedge setting.  Noting first that the doubling trick works with (uniform) Hedge in the usual set-up.  However, if we generalize Hedge to the non-uniform case so that the a-priori probability of each expert is not $p_1 = \cdots = p_n= 1/n$ but instead each expert depends on non-uniform a-priori probabilities $p_i$, we obtain a (non-uniform) regret bound for Hedge of $E[L_A] - L_i \le \sqrt{\log(1/p_i) T}$ which holds for all $i$ if we had foreknowledge of the optimum $\eta$.  But it does not seem likely we can get such a bound from the doubling trick as it is no longer obvious when to "double", i.e., the threshold of when to double depends now on a $log(1/p_i)$ rather than a $\log n$ term but we do not know the optimum $i$ in advance.  By analogy this is essentially the same case we are in with "tracking" with or without memory as we can view tracking algorithms as "meta-expert" algorithms where the a-priori probability of the meta-expert is not uniform but depends on $T,k,m$.  There are many other techniques to address the tuning of $\eta$ in the Hedge setting but to the best of our attempts they also do not transfer over to the non-uniform setting.  We list some other current modern references for tuning $\eta$ and beyond in the hedge setting.
>
> [1] A. Chernov and V. Vovk. Prediction with advice of unknown number of experts. In Proceedings of the Twenty-Sixth Conference on Uncertainty in Artificial Intelligence (UAI'10), pp. 117–125, 2010.
>
> [2] D. J. Foster, S. Kale, M. Mohri, and K. Sridharan. Parameter-Free Online Learning via Model Selection. In Advances in Neural Information Processing Systems, 2017.
>
> [3] K. Chaudhuri, Y. Freund, and DJ Hsu. A parameter-free hedging algorithm. In Advances in neural information processing systems, pp. 297-305, 2009.
>
>
> > *I agree that the contribution is mostly theoretical but some simulations could illustrate if there are practical differences between the two schemes (projections vs MPP) and if this factor 2 really makes a difference in real life.*
>
> We would ideally include simulations but due to the number of theoretical results in the paper we could not incorporate them due to space constraints. We are particularly interested in the application of our projection method to the problem of online portfolio selection with transaction costs (as discussed in e.g., lines 259-264). This research is ongoing. Note that simulations comparing the "memory" case to the "non-memory" case exist in [4, Section 6] which we will reference in Section 5 (Discussion).
>
> > *Is the regret bound also always better than the one of fixed-share when m<n? A figure similar to Figure 1 with n instead of k might be interesting.*
>
> The regret bound has an additional O((k-m)\log{(T/k)}) over the bound of Fixed Share, and thus is better when $n>T/k$. In line 134 we say that the bound improves on Fixed Share for all $m\leq k$ under mild assumptions on $n$, however in the final version of the paper we plan to add an explicit discussion or say "for sufficiently large n" instead.
>
> > *I didn't find the last paragraph before section 2.1 that motivates the use of projection very clear. I understand that there is more detail later but at that point, I had no idea why these examples motivated the use of projection.*
>
> Thank you for your feedback on this. We will add more detail to this paragraph to improve clarity in the text.
>
> To be clear, the primary intuition behind this motivating example of applying these algorithms to online portfolio selection is the following.
> If we use the loss function described on line 143, then each expert represents an asset/stock in a market, and the algorithm's weights reflects our (normalized) portfolio. The price changes in the market after each discrete trading period (e.g., each day) then update our portfolio weights automatically by naturally inducing the "loss update" (eq. 7 in Algorithms 1&2). That is, at this point we do not need to actively trade to re-balance. The secondary updates (projection or weight-sharing) change our weights which then require us to actively buy/sell to re-balance our portfolio to reflect these weights. In practice we must then pay transaction costs proportional to the amount bought or sold. Since Theorem 4 guarantees that our projection update is always more conservative than weight-sharing, then we gain an advantage in paying less in transaction costs while having the same regret bound guarantee.
>
> This is a significant result since many algorithms designed for online portfolio selection assume zero transaction costs and perform poorly in practice when transaction costs are included (see [30] for a survey).
>
> > *What is the advantage of using (c,1/c)-realizability with respect to exp-concavity? The latter seems more common.*
>
> Exp-concavity implies $(c,1/c)$-realizability, however observe that by assuming $(c,1/c)$-realizability we may use a prediction function other than the mean which in some cases gives tighter bounds. For instance with the square loss predicting with the mean gives $c=2$, however a more sophisticated prediction function gives a tighter bound with $c=1/2$ (see e.g., [22, Section 4]).
>
> > *State that (3) is for a well-chosen alpha.*
>
> Thank you for pointing this out. We will add a comment giving the optimal alpha ($\frac{k}{T-1}$) and cite [22] on line 84.
>
> > *line 70: square loss has c=1/2 only if the prediction and output domains are [0,1]*
>
> Thank you for pointing this out. This is an oversight on our part, we will add a comment on line 70 clarifying the domains to be $[0,1]$.

---

> > ### Comment · Reviewer_AJRz · 2021-08-19
> > **Quick response to authors from reviewer AJRz**
> >
> > Since I am travelling with very limited internet access, I only answer briefly to the authors. I read their response to my review and other reviews and I think that the authors did a good job answering most of my comments. The only point which is not adressed concerns the critic that this contribution is quite specialized and thus limited since it is of interest only for very few people. I decide to keep my score of 6.

---

### Official Review · Reviewer_t33Y · 2021-07-17

**Rating:** 7
**Confidence:** 4

**Summary:**

The paper give two improved algorithm for the expert advice setting with long term memory guarantees.
Both achieve a regret bound that is slightly better than the best previous algorithm given in [26].
The first algorithm is of the Mixing Past Posterior type and is particularly simple.
The second uses a generalized relative entropy projection algorithm that is also interesting.

**Ethical Concerns:**

N/a.

**Limitations And Societal Impact:**

Basic research. No direct societal impact.

**Main Review:**

+ Well written paper. Lots of intuition given throughout. Some further suggestions are given below.
Definitely interesting because the simple MPP version achieves the improved bound.
The projection version of the algorithm gives further intuition why the new MPP algorithm works.
- A bit specialized and it could be argued that the improvements are not significant enough.
However this is chasing an NP-complete problem. May be complementing the paper with NP hardness results
and exactly quantifying the gap would be helpful.

Overall rating: 6.5


Detailed comments:

Line 5 of intro: Add "the" before "learner".

Page 2, line -7: No! I don't think that generalizing this to the Hedge setting is trivial, because
each segment of a partition requires a different optimized learning rate that depend on
the loss of the best in that segment.
Adjusting Hedge to the long-term memory setting is thus challenging it seems already for the Fixed Share Setting.

Page 3, before (3): Give the alpha that achieves (3).

Page 3, after (7): Which gamma achieves the better bound and how much is the improvment?

Page 5, section 3: Projections also have been used to cap weights from above instead of below.

Page 7, line 7 of alg. 1/2: the dot over w is too small. Suggest to use a hat.

Section 3: You use a worst-case linear median algorithm. Those algorithms are complex divide and conquer
algorithm that have a high constant.
But there is a simple randomized median finding algorithm with average time complexity O(n) that is
quite practical! Suggest to mention that algorithm as well.

Also I suggest to look at this paper:
Efficient Projections onto the L1-Ball for Learning in High Dimensions by Duchi, Shalev-Shwartz, Singer and Chandra.

They do Euclidean projections, but the don't have the worst case O(n) algorithm it seems. Your techniques will probably lead to
an O(n) w.c. algorithm as well for the Euclidean projection case when the ranges are non-uniform.

Page 8, line 255: When k>>m, then the additional cost per switch is again c ln T/k.


**Time Spent Reviewing:**

5 hours

---

> ### Author Response · Authors · 2021-08-10
> **Response to Reviewer t33Y**
>
> Thank you for your review and comments. Please find our specific responses to your comments below:
>
> > *Line 5 of intro: Add "the" before "learner".*
>
> Thank you for your suggestion, we agree that this would help readability and we will make this change.
>
> > *May be complementing the paper with NP hardness results and (2) exactly quantifying the gap would be helpful.*
>
> We will reference the NP-hardness result of [4, Theorem 10] which in essence reduces three-dimensional matching to determining if their is an expert sequence of $k$ shifts from a sub-pool of $m$ experts from $n$ that produce binary prediction over the $T$ trials that match the $T$ labels with zero loss.
>
> We confess we are not exactly sure what you mean by "exactly quantifying the gap", although we can speculate it would be better if you clarify what is intended.
>
> > *Page 2, line -7: No! I don't think that generalizing this to the Hedge setting is trivial, because each segment of a partition requires a different optimized learning rate that depend on the loss of the best in that segment. Adjusting Hedge to the long-term memory setting is thus challenging it seems already for the Fixed Share Setting.*
>
> We respectfully disagree.  Foreknowledge of a single "global" $\eta$ suffices. Hedge has already been generalized to this setting see [20, Theorem 1 (set s=1)], but note that in their result the theorem does have the loss restricted to {0,1}. Additionally however, see [20, Theorem 14] which works for losses in [0,1]. In [34, Theorem 9] there is also a different generalization of the model (number of experts can increase over time) to the Hedge setting though they do not the tune learning rate in presenting their results. Finally note that absolute loss for the variable-share algorithm was considered in [22, Theorem 5].
>
> We, however, agree that if we do not assume optimal foreknowledge of the learning rate of $\eta$ then the problem is difficult.
>
> > *Page 3, before (3): Give the alpha that achieves (3).*
>
> Thank you for pointing this out. We will add a comment giving the optimal alpha ($\frac{k}{T-1}$) and cite [22] on line 84.
>
> > *Page 3, after (7): Which gamma achieves the better bound and how much is the improvment?*
>
> The $\gamma$ used to parameterize the power law in Equation (6) is separate to $\gamma^{t+1}_{q}$ defining the MPP mixing scheme (this is a poor choice of notation on our part, we will change the power law parameter to a different variable in the final version of the paper).
> The choice of $\gamma=1$ is taken from [4, Corollary 9], where the choice of $1$ allows a good bound to be proved in [4, Appendix B]. It is not immediately clear how different choices of $\gamma$ can improve the bound for this mixing scheme. We note that in [4, Appendix D] the authors consider a modified version of their mixing scheme with a tuned parameter $\gamma\neq 1$ which gives slightly better bounds. We do not consider this scheme however as it implicitly assumes $m$ is less than $\sim k/3$ whereas we consider all values of $m$ up to $k+1$.
>
>
> > *Page 5, section 3: Projections also have been used to cap weights from above instead of below.*
>
> Thank you. This is something we would like to add to the discussion, we are aware of capping weights from above in [23], but are not aware of any other instances for relative entropy projection.
>
> > *Page 7, line 7 of alg. 1/2: the dot over w is too small. Suggest to use a hat.*
>
> Thank you for pointing this out. We agree your suggestion will improve readability. We will make this change upon acceptance of the paper.
>
> > *Section 3: You use a worst-case linear median algorithm. Those algorithms are complex divide and conquer algorithm that have a high constant. But there is a simple randomized median finding algorithm with average time complexity O(n) that is quite practical! Suggest to mention that algorithm as well.*
>
> Thank you for pointing this out. This is a very good point and we would gladly add extra discussion in the paper in line 548, citing Floyd, R. & Rivest, R., 1975. Expected time bounds for selection. Communications of the ACM, 18(3), pp.165–172.
>
> > *Also I suggest to look at this paper: Efficient Projections onto the L1-Ball for Learning in High Dimensions by Duchi, Shalev-Shwartz, Singer and Chandra.
> They do Euclidean projections, but the don't have the worst case O(n) algorithm it seems. Your techniques will probably lead to an O(n) w.c. algorithm as well for the Euclidean projection case when the ranges are non-uniform.*
>
> Thank you! We will check if our methods can be generalized to the Euclidean case.
>
> > *Page 8, line 255: When k>>m, then the additional cost per switch is again c ln T/k.*
>
> Yes this is correct. Our bound improves on the bound (10) from [26] by not paying an additional cost of $c\ln{\frac{T}{k}}$ on the first time each expert is learned. Thus when $m\ll k$ the improvement is small, however the improvement is most impactful when memory is only required for some experts (see Figure 1).

---

### Official Review · Reviewer_gqjS · 2021-07-30

**Rating:** 7
**Confidence:** 5

**Summary:**

The paper studies improving regret bounds for memory tracking experts.  There are n experts for a learning task of T examples, out of which only m << n experts are useful.  The examples switch between different populations of examples k << T times, and are assumed to stay a sufficient time in each population segment.  Since m << k, experts are revisited, and the goal is to take advantage of the knowledge that experts are reused to reduce the regret.

The paper proposes following on prior work to learn weights for the different experts, and use non-fixed generalized share updates to share portions of weights assigned to an expert with other experts at subsequent rounds.  The amount of sharing is key to reducing regret.  Instead of directly following a generalized share update path, the approach update shares by the prediction quality of the expert, and then projecting the sharing weight vector to a subspace of the probability simplex such that shares do not fall below a dynamically updated threshold.

Upper bounds, claimed to be stronger than previous works, are then derived for the regret of the method, specifically lowering an extra O(log(T/k)) term per switch that is present in previous bounds.  This term is lowered to include only k - m terms instead of k, eliminating the cost of the first occurrence of each expert (with some second order correction).

The paper then goes on to show that the approach used can actually be mapped to a slightly different share update method but with the same regret upper bounds, which is a special case of the generalized share update rule.  The latter is then shown to also be a special case of a Bayesian mixing scheme that uses a geometric distribution.  It concludes that because of that, the bounds in this paper also apply to the Bayesian approach in Koolen et. al. 2012.

In addition, the paper presents a low complexity projection method from an unconstrained probability simplex to one in which the probabilities are constrained from below, to be used for the projection step of the proposed projection algorithm.  It then asserts that if adapting weights as proposed for mixing the experts’ advice costs proportionally to the L1 distance between the weights before and after the adaptation, the projection method is advantageous to the other derived approaches.


**Main Review:**

The paper addresses an interesting problem and is nicely written. The introduction gives a very good overview of the problem area, and positions the paper very well for the technical part.

While I would like to see this work published, if correct, I have several concerns about the actual results and their correctness.  I will appreciate it if the authors can address and clarify these comments, before I can recommend publication.  There are several other major points that I would like to see corrected before recommending for publication.  I am happy to iterate with the authors during the open review process to verify these comments are addressed.  Given these current concerns, I give a do not accept score.  However, if these issues are addressed, I’ll be happy to improve my score.

Theorem 3: It is not clear to me at this point that the proof covers all situations. For example, suppose we have m experts out of which only one is active for the first O(T) rounds.  All the remaining m-1 experts achieve very large losses during those rounds and also, each of these m-1 experts incurs very large loss on examples that another expert in the set achieves low loss.  Then, only in the last examples constituting the remaining k segments, the m-1 remaining experts become active.  It seems that in such a case each of the m-1 remaining experts will start with a weight that is (1-theta)^{O(T)} x alpha/n.  Since theta = O(k/T), wouldn’t this yield a term of O(km) in the bound, which, assuming m is omega(1), is larger than the order of the bound?  (This will not change if m << k, and is independent of whether some of these experts will reoccur in the last k segments).

Theorem 4: The paper concludes that the projection algorithm is better for minimizing rebalancing costs due to this theorem.  Is that a direct result of the lower limits not moving?  However, in the projection algorithm they do actually move, and then Theorem 4 may not apply to the algorithm.

The paper should state when it is at all useful to use the proposed approach earlier and more clearly.  While the combinatorial bound shows advantages over fixed share whenever m < n, the extra O((k-m) log(T/k)) regret of the proposed method makes it clearly inferior to fixed share if T/k > n and m = o(k).  Figure 1 does not cover cases like this.

Does the heuristic in the paragraph starting in line 199 really guarantee minimization of the divergence?  This should be proved.

The paper does not mention a large body of literature from information theory that is directly related to the problem of tracking experts, and presents bounds on cost of switches as well as algorithms that are very related to those of expert tracking.  These papers should be cited and mentioned in the presentation of the problem.  Below is a partial list of relevant papers (the first and the last 2 are specifically relevant to recurring experts).  There are additional references in the listed papers below that are relevant.

Y. M. Shtarkov, “Switching discrete sources and its universal encoding,” Probl. Inform. Transm., vol. 28, no. 3, pp. 95–111, July-Sept. 1992

N. Merhav, “On the minimum description length principle for sources with piecewise constant parameters,” IEEE Trans. Inform. Theory, vol. 39, pp. 1962–1967, Nov. 1993.

F. M. J. Willems, “Coding for a binary independent piecewise-identically-distributed source,” IEEE Trans. Inform. Theory, vol. 42, pp. 2210–2217, Nov. 1996

G. I. Shamir and N. Merhav, “Low complexity sequential lossless coding for piecewise stationary memoryless sources,” IEEE Trans. Inform. Theory, vol. 45, pp. 1498–1519, July 1999

G. I. Shamir, “On Asymptotically Optimal Sequential Lossless Coding for Memoryless
Switching Sources”, ISIT 2002.

G. I. Shamir and D. J. Costello, “Universal lossless coding for sources with repeating statistics”, IEEE Trans. Inform. Theory, vol. 50, pp. 1620-1634, Aug. 2004

Minor issues:

For this approach (and the ones it is based on) to work, one needs to know T, k, m in advance - this should be mentioned in the introduction.

Line 129 - the paper implies that the m log n term is paid to remember, not to learn.  However, this seems inconsistent with the combinatorial bound, where this term comes from choosing which m out of the n experts are the ones whose advice are used.

Line 150 - The idea mentioned here was first introduced in Shtarkov’s paper above.

There is use of v^t and tilde{v}^t - it is not clear how they differ.

To introduce PoDS in Section 3, it would help to add one intuition sentence saying something like: the algorithm projects the weights such that they don’t go below a certain threshold that was learned for each expert.

It would be better to change the order of Theorems 1 and 3.  First, Alg. 1&2 should be presented, stating the use of the projection result described later.  Then, Theorem 3 should be stated.  Then, Alg. 3 should be discussed and then Theorem 3.  The current structure is confusing and breaks the flow for a reader.  Also, Theorem 3 is the more important one for the results, and the flow of the paper should lead to it before talking about supplementary results like Theorem 1.

Line 210 - it would be better to use a different character from k, that is already used for the number of switches.


**Time Spent Reviewing:**

5

---

> ### Author Response · Authors · 2021-08-10
> **Response to Reviewer gqjS**
>
> Thank you for your review and comments. Please find our specific responses to your comments below:
>
> > *Theorem 3: It is not clear to me at this point that the proof covers all situations. For example, suppose we have m experts out of which only one is active for the first O(T) rounds. All the remaining m-1 experts achieve very large losses during those rounds and also, each of these m-1 experts incurs very large loss on examples that another expert in the set achieves low loss. Then, only in the last examples constituting the remaining k segments, the m-1 remaining experts become active. It seems that in such a case each of the m-1 remaining experts will start with a weight that is (1-theta)^{O(T)} x alpha/n. Since theta = O(k/T), wouldn’t this yield a term of O(km) in the bound, which, assuming m is omega(1), is larger than the order of the bound? (This will not change if m << k, and is independent of whether some of these experts will reoccur in the last k segments).*
>
> Theorem 3 gives a worst-case regret bound which holds for all possible data sequences. Note that $\theta=O(\frac{k}{mT})$ and thus in your example there is a factor of $m$ omitted from the denominator of $\theta$. Including this factor of $m$ in $\theta$ in your example yields a term of $O(k)$, not $O(km)$. Additionally note that if correct your argument would also apply to previous results (e.g., [26]), where "$\theta$"$=P_{sw}=\frac{k}{(m-1)(T-1)}$ (line 307).
>
> > *Theorem 4: The paper concludes that the projection algorithm is better for minimizing rebalancing costs due to this theorem. Is that a direct result of the lower limits not moving?*
>
> Theorem 4 does not depend on lower limits not moving, indeed for PoDS-$\theta$ they do move. The intuition is that the projection update is the most "conservative" update (w.r.t changing weights) we can do while still satisfying the lower limits on weights required for proving the regret bounds, whereas weight-sharing is more "aggressive". Additionally consider the case on a given trial where all lower limits are satisfied, then the projection will not "update" the weights whereas the weight-sharing update will almost always update the weights\*.
> Theorem 4 does apply to the algorithms given in the paper, under the assumption of "all else being equal" i.e., $\beta = \alpha v$ as stated in the theorem (line 272).
>
> \* In the interest of furthering this discussion we would like to point out a small correction we must make to Theorem 4: strict inequality holds only if $v\neq w$ (i.e., the RHS is non-zero), otherwise both norms are zero and equal. Of course this does not affect the overall result. We will add this to the theorem statement in the final version.
>
>
>
> > *The paper should state when it is at all useful to use the proposed approach earlier and more clearly. While the combinatorial bound shows advantages over fixed share whenever m < n, the extra O((k-m) log(T/k)) regret of the proposed method makes it clearly inferior to fixed share if T/k > n and m = o(k). Figure 1 does not cover cases like this.*
>
> Thank you, this is correct. We say that the bound improves on Fixed Share for all $m\leq k$ under mild assumptions on $n$ on line 134. In the final version of the paper we will add an explicit discussion or say "for sufficiently large n" instead.
>
>
> > *Does the heuristic in the paragraph starting in line 199 really guarantee minimization of the divergence? This should be proved.*
>
> The method described in the paragraph starting in line 199 is not a heuristic. The method does guarantee the minimization of the divergence and is proved in Appendix A (Proof of Theorem 1) in the supplementary material.
>
> In lines 484-505 we prove that the components of the projected vector either take on their lower limits, or are scaled by a constant.
>
> In lines 507-518 (Claim 6) we prove that components which take on their lower limits must have smaller ratios (of weights to limits) than any components which are scaled.
>
> In lines 519-522 (Claim 7) we prove that the divergence is minimized by setting the smallest number of components to their lower limits and scaling the remaining components such that they sum to $1$ and all limits are satisfied.
>
> We are happy to add an additional comment to line 199 stating that this method is an intuitive description of Algorithm 3 and is proved in Appendix A (we mention this in line 220 however we agree this should be introduced beforehand).
>
> > *The paper does not mention a large body of literature from information theory that is directly related to the problem of tracking experts, and presents bounds on cost of switches as well as algorithms that are very related to those of expert tracking. These papers should be cited and mentioned in the presentation of the problem. Below is a partial list of relevant papers (the first and the last 2 are specifically relevant to recurring experts). There are additional references in the listed papers below that are relevant.*
>
> Apologies, for missing these references from the universal coding literature.  We will cite and discuss these in the paper as well as make the necessary connections in the papers that they themselves cite.
>
> Note: although we can find some accounts of the Y. M. Shtarkov reference, we have so far been unable to determine if there is an English language translation of that reference which if you are aware of one we would be glad to be pointed in that direction.
>
>
> > *For this approach (and the ones it is based on) to work, one needs to know T, k, m in advance - this should be mentioned in the introduction.*
>
> Yes, we agree and we will do so.  Currently we have a short discussion of this in lines 324-328.  We note that there is in fact an extensive literature on "auto-tuning" "tracking the best expert (without memory)"  so far we have only cited the earliest reference that we are aware of [39].  More generally, auto-tuning is a much simpler problem for $(c,1/c)$-realizable losses as the optimal learning rate $\eta$ is fixed and independent of $T,k,m$ rather than say for example the absolute loss (Hedge setting).  One technique available for $(c,1/c)$-realizable losses is to have a prior over the parameters and then "mix" over all parameter setting.  This can be done with varying degrees of efficiency at a trade-off between computational complexity and regret.
>
> In fact [26, p.6] claims such a strategy is possible at no asymptotic computational cost and provides a regret bound.  We did not cite the result because it is provided without proof.  However, if we can verify the result we will then cite it.
>
>
> > *Line 129 - the paper implies that the m log n term is paid to remember, not to learn. However, this seems inconsistent with the combinatorial bound, where this term comes from choosing which m out of the n experts are the ones whose advice are used.*
>
> The $m \log n$ term is not paid to remember, but rather paid to learn the m unique experts, as the combinatorial bound suggests. We then pay $\sim\ln{T/k}$ per switch and an additional $\sim\ln{Tm/k}$ to "remember" an expert (line 253).
>
> > *Line 150 - The idea mentioned here was first introduced in Shtarkov’s paper above.*
>
> In the full paper we will clarify the issue of priority.  Note, however, the 1994 reference to the "The Weighted Majority Algorithm" builds on two early technical reports UCSC-CRL-89-16, UCSC-CRL-92-14, whose content is sketched as an "Abstract" in FOCS 1989.
>
> So we propose to clarify the text around line 150 to be something similar to
> "Switching (without memory) in online learning was first introduced in [32] (see also [FOCS89] and independently in the context of universal coding in [Shtarkov92])."
>
> > *There is use of v^t and tilde{v}^t - it is not clear how they differ.*
>
> Thank you for pointing this out. To clarify, $\tilde{v}^t$ refers to any "generalized" share update (8), whereas $v^t$ refers to Share-$\theta$ only (17). Upon reflection we agree the use of tilde is not necessary since the discussion of generalized share update precedes Share-$\theta$. This will therefore be removed in the final version.
>
> > *To introduce PoDS in Section 3, it would help to add one intuition sentence saying something like: the algorithm projects the weights such that they don’t go below a certain threshold that was learned for each expert.*
>
> Thank you very much for this suggestion. We agree and would like to add an intuitive sentence after line 169.
>
> > *It would be better to change the order of Theorems 1 and 3. First, Alg. 1&2 should be presented, stating the use of the projection result described later. Then, Theorem 3 should be stated. Then, Alg. 3 should be discussed and then Theorem 3. The current structure is confusing and breaks the flow for a reader. Also, Theorem 3 is the more important one for the results, and the flow of the paper should lead to it before talking about supplementary results like Theorem 1.*
>
> Thank you very much for this suggestion. We absolutely agree! To improve the flow of the paper we will move section 3.1 (lines 192-223) to just before line 259 upon acceptance.
>
> > *Line 210 - it would be better to use a different character from k, that is already used for the number of switches.*
>
> Thank you, we agree. This will be changed in the final version also.

---

> > ### Comment · Reviewer_gqjS · 2021-08-12
> > **Quick Response to Authors from Reviewer gqjS**
> >
> > I thank the authors for their detailed response to my review.  While I am currently in a very tight time schedule, and unable to thoroughly respond to the rebuttal, and I'll be away from office in the coming 1.5 weeks, I'd like for now to give a high level response, and if necessary I'll add another response later.
> >
> > I think the authors did a nice job responding my comments.  My main concern was with Theorem 3, due to my oversight of missing the m term in theta.  I believe that the response by the authors covers this and addresses my main concern, and I'm raising my score given this response.  I would still like to review the other technical claims, but I'll have to do that later.
> >
> > I believe that the Shtarkov paper was translated to English, and I believe I had access to a printed copy very long ago, but I doubt I can find that.  If I do, I'll try to pass it on to the authors through the area chair.

---

> ### Comment · Reviewer_t33Y · 2021-08-19
> **Thanks for the thorough response and to all the other reviewer for their detailed and thoughtful reviews.**
>
> Final comment: Yes - the results are pretty specialized. But I stick to my score.
> We encourage to add experimental comparisons of the algorithms to the appendix.

---

### Decision · Program_Chairs · 2021-09-28

**Decision:**

Accept (Poster)

**Comment:**

The reviewers have agreed that the presented results are novel, the paper is nicely written, and contains good ideas. On the other hand, the paper is probably only interesting to a small part of the community, providing a little improvement to a well-studied problem, and the algorithm also needs to know the problem parameters in advance. Nevertheless, the positives outweigh the negatives, hence I recommend acceptance.

**Consistency Experiment:**

NeurIPS has a long history of experimentation. In 2014, NeurIPS ran an experiment in which 10% of submissions were reviewed by two independent committees to quantify the randomness in the review process. This year, we repeated a variant of this experiment to see how the quality of the review process has changed over time.  This paper was part of the experiment and was therefore assigned to two committees (consisting of reviewers, an Area Chair, and a Senior Area Chair) that reached independent decisions.  If both committees made the same recommendation, this recommendation was followed. If a single committee recommended acceptance, the paper was accepted (with the exception of a few cases in which the other committee identified what we considered a fatal flaw, e.g., an error in a key result).

Both committees reached the same decision: **Accept (Poster)**

The other committee assigned to the paper recommended **Accept (Poster)**.  You can find the other set of reviews, along with any follow up discussion with the authors here:
https://openreview.net/forum?id=x_sdq4ZYSOl